# CAN TEST-TIME COMPUTATION MITIGATE REPRODUCTION BIAS IN NEURAL SYMBOLIC REGRESSION?

## ABSTRACT

Symbolic regression aims to discover mathematical equations that fit given numerical data. It has been applied in various fields of scientific research, such as producing human-readable expressions that explain physical phenomena. Recently, Neural symbolic regression (NSR) methods that involve Transformers pre-trained on large-scale synthetic datasets have gained attention. While these methods offer advantages such as short inference time, they suffer from low performance, particularly when the number of input variables is large. In this study, we analyze the reasons for this limitation and suggest ways to improve NSR. We first provide a theoretical analysis showing that, under naive inference strategies, Transformers are unable to construct expressions in a compositional manner while verifying their numerical validity. Next, we explore how Transformers generate expressions in practice despite the lack of compositional generalizability. Our empirical analysis shows that the search space of NSR methods are greatly restricted due to reproduction bias, where the majority of generated expressions are merely copied from the training data. We finally examined if tailoring test-time strategies can reduce reproduction bias and improve numerical accuracy. We empirically demonstrate that providing additional information to the model at test time can significantly mitigate reproduction bias. On the other hand, we also found that reducing reproduction bias does not necessarily correlate with improved accuracy. These findings contribute to a deeper understanding of the limitation of NSR approaches and offer a foundation for designing more robust, generalizable symbolic regression methods.

## 1 INTRODUCTION

Discovering underlying equations from collected experimental data is a crucial process in many fields of scientific research. Symbolic regression is a branch of regression analysis that seeks to automatically identify underlying mathematical expressions. In contrast to methods that model data without explicit mathematical expressions, symbolic regression offers advantages in terms of interpretability and generalizability. This is because the outputs of symbolic regression are usually compact, human-readable equations, making them less susceptible to overfitting. However, symbolic regression is a challenging task due to its vast search space; the number of possible mathematical expressions grows exponentially with expression length or the number of input variables. Applications for symbolic regression span various fields of scientific research such as physics (Tenachi et al., 2023), materials science (Wang et al., 2019), and weather forecasting (Abdellaoui & Mehrkanoon, 2021).

Various methods for symbolic regression have been proposed in recent years. Traditionally, approaches based on genetic programming (GP) (Koza, 1994) have been employed to solve symbolic regression. These methods tend to be computationally expensive because they generate each expression entirely from scratch. To mitigate this inefficiency, a research direction called neural symbolic regression (NSR) has emerged. NSR methods leverage encoder-decoder Transformer architectures Vaswani et al. (2017) pre-trained on large-scale synthetic datasets Biggio et al. (2021); Valipour et al. (2021). NSR methods generate expressions similar to natural language processing tasks, where expressions are generated token-by-token in a auto-regressive manner. Since a single forward pass through the Transformer suffices to output a mathematical token (e.g., $x_1, \sin, +$), NSR models can generate solutions far more quickly than GP-based approaches. However, NSR methods often falls short in terms of numerical accuracy, with particularly poor performance when the number of input

variables is large (Kamienny et al., 2022; Bendinelli et al., 2023). This study aims to uncover the underlying cause of this drawback and explore methods to alleviate it.

Our analysis began with the question of what mechanisms a Transformer relies on to select the next token while generating an expression. We first conducted a theoretical analysis of the limitations faced by Transformers when generating mathematical expressions. An ideal way to generate the next token would be to generate the token that, if appended to the expression generated so far, most increases the probability for the final expression to fit the numerical data. However, by using circuit complexity theory, we show that Transformers fail to generate expressions in such ways; they cannot compositionally generate mathematical expressions while taking numerical data into account. For example, consider a situation where a Transformer has generated an expression up to $x_1^2 + \sin(x_2)+$. Our analysis implies that Transformers are unable to internally compute which leaf token (e.g., $x_1, x_2, x_3, \ldots$) would lead to an expression that best fits the input numerical data. The result indicates that in practice, Transformers generate expressions by some alternative mechanism instead of generating them in a compositional manner.

We next investigated how NSR methods generate expressions under empirical conditions. We hypothesized that, in NSR methods, naively using a Transformer for inference leads to **reproduction bias**, meaning that models struggle to generate novel expressions not seen during training and instead tend to generate expressions copied from the training data. Given that the expressions in the training data typically represent only a small subset of the full space of possible expressions, our hypothesis implies that standard NSR methods operate within a significantly constrained search space. We investigated this hypothesis in NSR methods such as NeSymReS (Biggio et al., 2021), a pioneering work in NSR models. We found that the majority of expressions generated by Transformers are expressions that were included in the training dataset, which supports our hypothesis of reproduction bias. Prior work has highlighted NSR methods' limited generalizability with respect to the range of numerical data—e.g., models trained on data whose input variable $x$ lies in the interval $[-1, 1]$ often fail when evaluated on inputs from the wider interval $[-2, 2]$ (Li et al., 2024; Shojaee et al., 2023). However, the reproduction bias that we identify is orthogonal to this phenomenon, and represents an even more fundamental limitation: NSR models often fail to generalize even within their training domain. This work is the first to show that standard NSR models primarily copy training expressions instead of composing familiar components into genuinely novel formulas.

Towards the end of this paper, we explore methodologies to mitigate the reproduction bias of standard NSR models and improve numerical accuracy. We focus particularly on test-time strategies and investigated how they affect reproduction bias and numerical accuracy. We compared three strategies: decoding with a large beam size, decoding using MCTS, and providing verification feedback at the subtree level. The last strategy is a new method that we propose, which we refer to as neural symbolic regression guided by verified subtrees (NSR-gvs). We found that providing new information to the model during test-time leads to generating expressions beyond the training dataset. However, we also identified cases where reproduction bias was mitigated but numerical accuracy decreased, as well as cases where numerical performance improved despite little alleviation in reproduction bias. We conclude this paper by discussing the underlying causes of these differences across varying types of test-time strategies.

The contributions of our work are summarized as follows:

- We conducted a theoretical analysis and formally show that Transformers lack the ability to compositionally generate expressions while accounting for numerical data.
- We empirically demonstrate, under various settings, that naively applying a Transformer to symbolic regression leads to reproduction bias.
- We compared varying test-time computing strategies and analyzed how such strategies affect reproduction bias and numerical accuracy.

## 2 RELATED WORK

Several approaches to symbolic regression exist, such as GP, brute force algorithms, reinforcement learning, and NSR. Since our study focuses on analyzing and improving NSR methods, we mainly describe NSR in detail in this section, and provide explanation for other symbolic regression methods in Section F.

Table 1: Comparison between our work and other major NSR studies

| NSR Methods | Automatic Training Data Generation | Direct Constant Prediction | Information Added During Test-time | Assessing Reproduction Bias |
|---|---|---|---|---|
| Biggio et al. (2021) | ✓ | - | - | - |
| Kamienny et al. (2022) | ✓ | ✓ | - | - |
| Shojaee et al. (2023) | ✓ | ✓ | MCTS Feedback | - |
| Li et al. (2024) | - | - | Historical Context | - |
| Bendinelli et al. (2023) | ✓ | - | Prior Knowledge | - |
| Ours (NSR-gvs) | ✓ | - | Verified Subtrees | ✓ |

Traditional symbolic regression methods such as GP generate each equation from scratch, resulting in long inference times, with equation generation potentially taking hours. In order to achieve a shorter inference time, studies such as NeSymReS (Biggio et al., 2021) and SymbolicGPT (Valipour et al., 2021) carried out large scale pre-training of Transformers. In these studies, an artificial dataset consisting of millions of randomly generated equations was used for training. These methods, often categorized as NSR, can generate an expression in just a few seconds, significantly reducing inference time compared with other approaches.

In the recent years, a number of studies, summarized in Table 1, have focused on enhancing NSR methods. Studies such as (Kamienny et al., 2022) and (Vastl et al., 2024) proposed an end-to-end approach using a Transformer model to directly predict full mathematical expressions including constants, whereas previous methods followed a two-step procedure where constant fitting had to be done separately. Lalande et al. (2023) analyzed several different architectures to find the suitable encoder architecture for NSR. Shojaee et al. (2023) focused on improving the decoding strategy for NSR, incorporating the MCTS algorithm during the generation of expressions. In their study, Li et al. (2024) trained a Transformer model to imitate the process of improving mathematical formulas, as performed in the reinforcement learning-based approach proposed by Mundhenk et al. (2021). Bendinelli et al. (2023) proposed a model called NSRwH that enables incorporating prior knowledge, which is often available during application in scientific research. For example, scientists may anticipate symmetries between variables or expect certain partial expressions to appear in the mathematical laws governing the data. NSRwH adds a dedicated encoder that processes such prior knowledge, allowing the model to generate expressions that are consistent with both the numerical data and the provided prior knowledge.

More recently, there has been growing research on methods that iteratively refine mathematical expressions, most of which rely on large language models (LLMs) rather than pre-trained Transformer models. These methods are similar to NSR-gvs, one of the test-time computation approaches considered in this paper, in that they iteratively improve their outputs by incorporating feedback from the generated expressions. In (Merler et al., 2024) and (Sharlin & Josephson, 2024), the authors introduce approaches in which a base equation structure is generated using LLMs, and the equation is subsequently improved iteratively by receiving feedback from external numerical solvers. While Shojaee et al. (2023) follows a similar methodology, it incorporates supplementary descriptions regarding the variables in the prompt, facilitating more effective use of the LLM's scientific knowledge. Grayeli et al. (2024) proposes a method that uses an LLM to identify "concepts" representing features of high-performing expressions and leverages them to further evolve a set of equations. Zhang et al. (2025) introduces an iterative algorithm that replaces features of suboptimal expressions at each step while incorporating relevant expressions as needed. Pre-training-free methods described above may have the potential to address some of the limitations of conventional NSR approaches. On the other hand, using a pre-trained Transformer, as opposed to an LLM, offers certain advantages similar to those of small language models (SLMs), such as keeping the model size manageable and enhancing domain specificity through careful design of the training data. For these reasons, we believe that conducting a deeper analysis of pre-trained Transformer-based NSR and exploring ways to improve it remains a valuable research direction.

## 3 PROBLEM FORMULATION

We formalize NSR as the problem of learning a parameterized symbolic regressor $S_{\boldsymbol{\theta}}$ that maps a numerical dataset $\mathcal{D}$ to a symbolic expression $\hat{e} = S_{\boldsymbol{\theta}}(\mathcal{D})$. The learning algorithm is formulated as minimizing a loss that measures how well $\hat{e}$ matches the ground-truth expression $e^*$ underlying $\mathcal{D}$. In this section, we specify how synthetic training pairs $(e^*, \mathcal{D})$ are generated in NeSymReS (Biggio et al., 2021), since it is the foundational work underlying our research.

### 3.1 SYNTHETIC EXPRESSION DISTRIBUTION

We first sample a random binary–unary tree whose internal nodes are operators and whose leaves are variables.

Let $\mathcal{V} = \{x_1, \ldots, x_d\}$ be a finite set of variables, $\mathcal{O}_{\text{bin}}$ the binary operators (e.g., $\{+, -, \times, \div\}$), and $\mathcal{O}_{\text{un}}$ the unary operators (e.g., $\{\sin, \cos, \log, \exp\}$). Denote by $\mathcal{C} = [c_{\min}, c_{\max}] \subset \mathbb{R}$ the interval from which numeric constants are drawn. The complete vocabulary for the expression is $\Sigma = \mathcal{V} \cup \mathcal{O}_{\text{bin}} \cup \mathcal{O}_{\text{un}} \cup \mathcal{C}$.

Let $\mathcal{E}$ be an expression space and $p_{\mathcal{E}}$ be the generator of symbolic expressions employed in NeSymReS (Biggio et al., 2021). We also denote by $p_{\text{Tree}}$ the generator of unary-binary trees introduced by Lample & Charton (2019). We write $e^* \sim p_{\mathcal{E}}$ for the following procedure.

1. Draw a random binary-unary tree $T \sim p_{\text{Tree}}$.

2. Assign internal nodes independently and uniformly from $\mathcal{O}_{\text{bin}} \cup \mathcal{O}_{\text{un}}$, and leaves uniformly from the variable set $\mathcal{V}$, resulting in a template expression $e_{\text{templ}}$.

3. For each unary operator $u$, sample a constant $c_{\text{mul}}$ from distribution $\mathcal{D}_{\text{mul}}$ and replace $u$ with $c_{\text{mul}}u$; otherwise keep the unary operator as is.

4. For each variable $x$, sample a constant $c_{\text{mul}}$ from distribution $\mathcal{D}_{\text{mul}}$ and a constant $c_{\text{add}}$ from distribution $\mathcal{D}_{\text{add}}$ and replace $x$ with $c_{\text{mul}}x + c_{\text{add}}$; otherwise keep the variable as is.

5. The resulting expression is the final $e^* \in \mathcal{E}$.

### 3.2 SYNTHETIC DATASET GENERATION

Given an expression $e^*$, we construct the dataset

$$\mathcal{D} = \{(\mathbf{x}_i, y_i)\}_{i=1}^n, \qquad x_{ij} \sim \mathcal{U}([x_{\min,j}, x_{\max,j}]) \quad \text{for} \quad j = 1, 2, \ldots, d, \qquad y_i = e^*(\mathbf{x}_i).$$

Where $\{[x_{min,j}, x_{max,j}]\}_{j=1}^d$ denotes the intervals for each independent variable. The joint distribution of training pairs is therefore $(e^*, \mathcal{D}) \sim p_{\mathcal{E}} \times \mathcal{G}$, where $\mathcal{G}$ denotes the above stochastic data generation process.

We now denote by $\Gamma = \mathcal{V} \cup \mathcal{O}_{\text{bin}} \cup \mathcal{O}_{\text{un}} \cup \{C, \text{END}\}$ the vocabulary for token sequences, where $C$ is a placeholder token to represent constants, and END denotes the explicit end-of-sequence marker. The vocabulary $\Gamma$ is slightly different from $\Sigma$ since continuous numeric constants cannot be represented with a finite number of tokens.

Let $\text{seq} : \mathcal{E} \longrightarrow \Gamma^*$ be a serialization map that converts any symbolic expression into its unique prefix token representation. For a ground–truth expression $e^* \in \mathcal{E}$ we set

$$\mathbf{s}^* = \text{seq}(e^*) = (s_1^*, \ldots, s_L^*), \qquad L := |\mathbf{s}^*|.$$

The predictive distribution $q_{\boldsymbol{\theta}}(\cdot \mid \mathbf{s}_{<j}, \mathcal{D})$ is realized by an encoder–decoder Transformer parametrized by $\boldsymbol{\theta}$. Conditioned on the dataset $\mathcal{D}$ (encoded by the encoder) and the previously emitted prefix $\mathbf{s}_{<j}$, the decoder outputs a probability over the next token $s_j \in \Gamma$.

The token-level loss for a single training pair $(e^*, \mathcal{D})$ is then

$$L_{\text{tok}}(e^*; \boldsymbol{\theta}) = -\sum_{j=1}^L \log q_{\boldsymbol{\theta}}(s_j^* \mid \mathbf{s}_{<j}^*, \mathcal{D}). \tag{1}$$

Note that, in practice, the training dataset is the collection of $e_{\text{templ}}$, and both $e^*$ and $\mathcal{D}$ are generated dynamically during training. Further details concerning the work of NeSymReS (Biggio et al., 2021) are described in Appendix A.

## 4 THEORETICAL ANALYSIS ON EXPRESSION GENERATION ABILITY OF TRANSFORMERS

Transformer-based symbolic regression tend to suffer from low performance, particularly when the number of input variables is large. In this section, we explore the theoretical basis of this limitation. Ideally, Transformers should be able to generate tokens in a compositional manner, while maximizing the probability for the final expression to fit the numerical data. However, we show that Transformers inherently lack the capacity to generate expressions compositionally while accounting for their numerical characteristics. We introduce a simplified version of the symbolic regression task and show that Transformers are not expressive enough to solve the task.

We define the last-token prediction problem as the task of predicting the most suitable final token in an otherwise complete mathematical expression. Although predicting the entire optimal expression is NP-hard Virgolin & Pissis (2022), this task is much easier since the search space is limited to several leaf tokens. We present a formal definition of this task in the following.

We first introduce $\text{expr} : \Gamma^* \times \mathbb{R}^{(d+1)*n} \longrightarrow \mathcal{E}$, a function that maps a token sequence $\mathbf{s}$ to the most appropriate expression $e_{\mathbf{s}}$ that can be represented by $\mathbf{s}$, taking numerical data $\mathcal{D}$ into account. Since the token sequence may contain the placeholder token $C$ representing constants, the mapping is tasked with identifying the optimal values for these constants and transforming the sequence into a corresponding expression tree.

**Definition 1** (Last-token prediction problem). *Given numerical data $\mathcal{D}$ of $n$ features-value pairs $(\mathbf{x}_i, y_i) \in \mathbb{R}^d \times \mathbb{R}$, a metric $\mathcal{L} : \mathbb{R}^n \times \mathbb{R}^n \to \mathbb{R}$, and an incomplete token sequence $\widetilde{\mathbf{s}}$ that forms a prefix representation of an expression when terminated with a leaf token, the last-token prediction problem asks for finding a leaf token $u^*$ such that:*

$$u^* = \operatorname*{argmin}_{u \in \Gamma} \mathcal{L}(\mathbf{y}, e_{(\widetilde{\mathbf{s}}, u)}(\mathbf{x})),$$

*where $e_{(\widetilde{\mathbf{s}}, u)} = \text{expr}((\widetilde{\mathbf{s}}, u), \mathcal{D})$ with $(\widetilde{\mathbf{s}}, u)$ representing the concatenation of sequence $\widetilde{\mathbf{s}}$ and token $u$. When the length of $\widetilde{\mathbf{s}} = m$, we denote this problem as $\text{LastTokenPrediction}(m)$.*

For the analysis, we assume a bounded-depth log-precision Transformer as in (Feng et al., 2023; Merrill & Sabharwal, 2023b;a; Strobl, 2023), a realistic setting where the intermediate computation values of the Transformer are limited to $O(\log k)$ bit precision, with $k$ denoting the maximal length of the input sequence. We now present the theoretical result stating that Transformers with bounded size cannot solve the last-token prediction problem.

**Theorem 1.** *Assume $\mathsf{TC}^0 \neq \mathsf{NC}^1$. For any integer $D$ and any polynomial $Q$, there exists a problem size $m$ such that no log-precision Transformer defined in Section E.1 with depth $D$ and hidden dimension $d \leq Q(m)$ can solve $\text{LastTokenPrediction}(m)$.*

We show the above theorem by leveraging circuit complexity theory. Specifically, $\mathsf{TC}^0$ and $\mathsf{NC}^1$ are types of circuit complexity classes, and it is generally conjectured that $\mathsf{TC}^0 \subsetneq \mathsf{NC}^1$. Prior work (Merrill & Sabharwal, 2023b) shows that log-precision Transformers can be simulated with $\mathsf{TC}^0$ circuits. We provide a proof for the above theorem by showing that the complexity of the last-token prediction problem is lower bounded by $\mathsf{NC}^1$. Detailed specifications of the problem setting and proof of the theorem are provided in Appendix E.

Although the final token of a mathematical expression is arguably the easiest to predict among its components, the above theorem shows that even this seemingly simple task presents substantial difficulties for Transformer models.

## 5 EXPLORING REPRODUCTION BIAS IN NSR

~~In Sec. 4, our theoretical analysis showed that Transformers lack the ability to generate expressions in a compositional manner while accounting for numerical data. Given the limitations~~

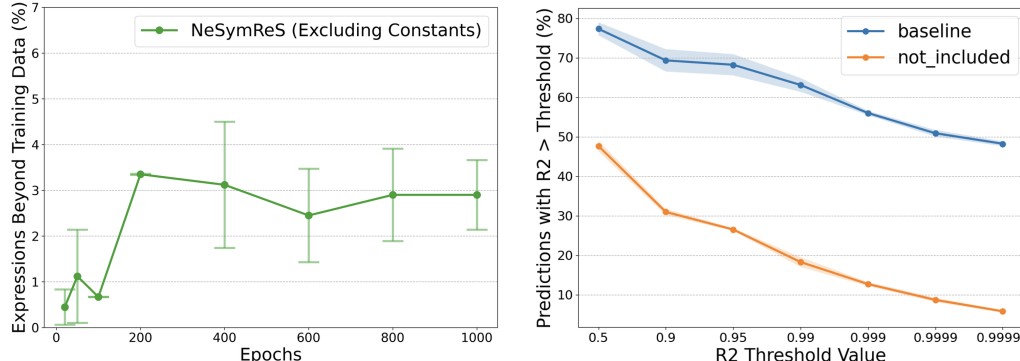

Figure 1: (Left) Percentage of expressions beyond the training dataset generated by NeSymReS on the `not_included` dataset. Throughout the training procedure, NeSymReS can hardly generate expressions that are not included in the training data, indicating strong reproduction bias. (Right) NeSymReS exhibits strong fitting performance on the `baseline` dataset but performs poorly on expressions from the `not_included` dataset, whose tree structures are absent from the training data. The result indicates the severe effect of reproduction bias on numerical accuracy.

~~of Transformers described above, this section empirically analyzes how expressions are actually generated by NSR models.~~ When generating expressions in an auto-regressive manner, a seemingly appropriate strategy would be to compositionally produce the next token that maximizes accuracy, conditioned on both the previously generated partial expression and the numerical data. However, our theoretical analysis from the last section showed that Transformers lack the ability to do so, bringing us to the following question: *How, in practice, does a transformer generate expressions during inference?* In this section, we empirically analyze how expressions are actually generated by NSR models. We demonstrate that NSR models primarily rely on reproduction—that is, they tend to generate expressions by directly copying those seen in the training data.

## 5.1 REPRODUCTION BIAS IN SIMPLIFIED SETTING

We first tested how expressions are generated in NeSymReS, which is the method that we mainly focus on in this study. We examined whether expressions generated by NeSymReS are merely copies from the training data or newly constructed formulas generated compositionally by the model.

We constructed a simplified training dataset consisting of 100K equations. The allowed operators were `add`, `sub`, `sin`, `cos`, `tan`, and `exp`, with up to 5 independent variables per equation. We then trained a NeSymReS model on this dataset for $1,000$ epochs. The variation of operators was limited due to the complicated training procedure of NeSymReS, where expressions with operators such as `mul` or `pow` are dynamically transformed and presented in different forms across epochs, making it difficult to judge whether the model's generated expressions are novel or memorized from training. The dataset size was also kept relatively small due to computational cost and to balance the size of the training data against the size of the search space.

As outlined in Section 3, the training dataset for NeSymReS comprises multiple instances of $e_{\text{templ}}$, each of which is an expression tree without numerical constants. Accordingly, we assessed the novelty of output expressions at the level of tree structure while ignoring numerical constants.

For evaluation, we constructed two test datasets: `not_included` and `baseline`, each containing 150 expressions. For the `not_included` set, we removed every $e_{\text{templ}}$ appearing in the training data. In contrast, the `baseline` set was sampled directly from the generator $p_{\mathcal{E}}$ without any filtering. We associated each expression with 100 data points, generated in the same way as during training. We set the beam size to 5 for this experiment.

To evaluate fitting performance, we used the $R^2$ score, defined as follows. Given a test equation, a set of $n$ data points $\{\mathbf{x}_i, y_i\}_{i=1}^n$, and the corresponding model predictions $\{\hat{y}_i\}_{i=1}^n$, the $R^2$ score is

computed as:

$$R^2 = 1 - \frac{\sum_{i=1}^{n} (y_i - \hat{y}_i)^2}{\sum_{i=1}^{n} (y_i - \bar{y})^2} \quad \text{where} \quad \bar{y} = \frac{1}{n} \sum_{i=1}^{n} y_i.$$

Note that these $m$ evaluation points are distinct from the inputs provided to the model at test time. By definition, $R^2 \leq 1$, and values closer to 1 indicate that the predicted outputs closely match the true equation. In our experiments, we counted the number of predictions whose $R^2$ exceeds thresholds of $0.5, 0.9, 0.95, 0.99, 0.999, 0.9999$, and $0.9999$, respectively. This allows us to assess the model's ability to fit the data under both moderate and stringent accuracy requirements.

Figure 1 shows the results for NeSymReS under the simplified setting. The left figure demonstrates NeSymReS's ability to generate novel expressions using the `not_included` dataset. As this dataset comprises instances of $e_{\text{templ}}$ unseen in the training data, the model is expected to produce previously unseen tree structures. However, the result indicates that NeSymReS struggles to generate expression trees beyond the training data across varying epochs. After 1000 epochs of training, over 97% of the generated expression trees were direct copies from the training data, which highlights the strong reproduction bias and reveals that the search space of NeSymReS is severely restricted. The right figure demonstrates how this reproduction bias negatively affects numerical accuracy, where NeSymReS's fitting performance on the `not_included` and `baseline` datasets are compared. The results indicate a substantial drop in performance for expressions whose tree structures are not present in the training data, compared with those sampled randomly. This suggests that for expressions not seen during training, the model's reproduction bias directly leads to poor numerical accuracy. This result also helps explain why NSR methods often fail to achieve high performance on expressions with many input variables; an increase in the number of input variables leads to an expanded search space, thereby increasing the likelihood that a given expression is absent from the training set.

### 5.2 REPRODUCTION BIAS IN PRACTICAL SETTING

Due to the complicated training procedure of NeSymReS, the above analysis was carried out in a simplified setting. To examine whether reproduction bias is a general phenomenon, we conducted an additional analysis in a more practical setting using transformer4sr (Lalande et al., 2023), a method similar to NeSymReS but with a simpler training process. In transformer4sr, no dynamic transformations of expressions are applied during training, which makes it much easier than in NeSymReS to verify whether the generated expressions are included in the training data. We were also able to analyze the novelty of the expressions not only in the tree-structure (excluding constants) level, but also taking into account the position of the constant placeholder tokens.

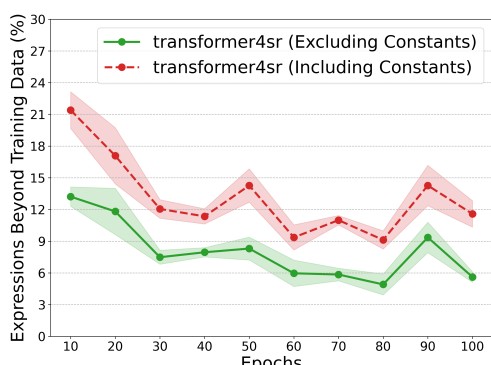

Figure 2: Reproduction Bias in transformer4sr under practical setting. Even for practical settings, the majority of generated expressions are expressions copied from the training data.

In this experiment, we followed the model architecture, training data size, number of epochs, operator selection, and inference strategies described in Lalande et al. (2023). We constructed a training dataset consisting of 1.5M equations and the model was trained for 100 epochs. We used the full set of operators, which are `add`, `mul`, `cos`, `log`, `exp`, `neg`, `inv`, `sqrt`, `sq` (squared), `cb` (cubed), and the number of independent variables were 6. We constructed a test set similar to `not_included` in the previous analysis, which consists of 300 expressions that were not included in the training data.

Figure 2 shows the result for transformer4sr's ability to generate expressions beyond the training data. After 100 epochs of training, less than 12% of the expressions generated by transformer4sr were novel expressions (taking into account the position of the constant placeholder tokens) beyond

the training data, and less than $6\%$ of the expressions had novel tree structures (excluding constants). The result demonstrates that reproduction bias persists even under more practical settings.

## 6 CAN TEST-TIME STRATEGIES MITIGATE REPRODUCTION BIAS?

The results from the previous section indicate that the search space of NeSymReS is mostly confined to expressions seen during training due to reproduction bias. Since our theoretical analysis indicates that naively performing next-token prediction makes it difficult to generate novel expressions in a compositional manner, we investigated the possibility of devising inference-time computational techniques to reduce reproduction bias in this section. Our hypothesis is that providing the model with hints about which tokens are appropriate could help steer the model to generate expressions that were not included in the training data. We begin by briefly introducing the three test-time strategies employed in our experiments. The detailed explanation for the strategies are presented in Section B.

### 6.1 TEST-TIME STRATEGIES

**Decoding with large beam size.**    Beam search serves as the default decoding strategy employed by NeSymReS. During decoding, Given a beam size of $b$, the decoding process generates $b$ candidate sequences via beam search. Each candidate's constant placeholders are subsequently optimized using the Broyden–Fletcher–Goldfarb–Shanno (BFGS) algorithm (Fletcher, 2000). The expression exhibiting the highest numerical accuracy on the test data is then selected as the model's output. While the experiments in Section 5.1 used a beam size of $b = 5$, in this section we conducted experiments with a larger beam size of $b = 150$. Since increasing the beam size does not provide the model with any additional information, our hypothesis is that simply adopting a decoding strategy with a larger beam size will not alleviate reproduction bias.

**Transformer-based planning for symbolic regression (TPSR).**    TPSR (Shojaee et al., 2023) is a method that leverages MCTS during decoding time. In TPSR, the process starts by preparing a pre-trained NSR model (e.g., the NeSymReS model). Instead of relying on standard decoding methods like beam search, the method generates tokens using MCTS, where both the expansion and evaluation stages of MCTS leverage the pre-trained NSR model. In the expansion phase, to avoid unnecessary exploration, the set of expandable tokens is restricted to the top-$k_{\max}$ candidates based on the logits from the NSR model. During the evaluation phase, the NSR model first completes the remainder of the expression following the expanded token. The completed expression is then evaluated primarily based on its fitting accuracy, with additional consideration given to its complexity. In the experiments presented in this section, we used the default hyperparameter settings of TPSR as specified in the original paper; we set the number of rollouts to $r = 3$, the number of expandable tokens to $k_{\max} = 3$, and beam size for expression completion to $b = 1$.

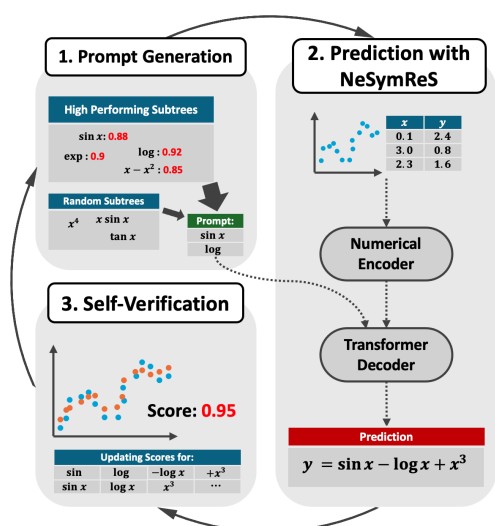

Figure 3: Overview of NSR-gvs's inference procedure. We first sampled subtrees from the candidate pool, then supplied them to the model together with numerical data. Then, the generated prediction is numerically verified and the self-verification feedback is used to update the candidate pool. This procedure is performed repeatedly to generate better predictions over time.

**NSR-gvs.**    TPSR provides feedback to the model by assigning a reward to each token, reflecting the quality or appropriateness of that token. In contrast, we hypothesized that incorporating feedback at the subtree level as well may have a positive effect on the model. To this end, we propose NSR-gvs, a method grounded in

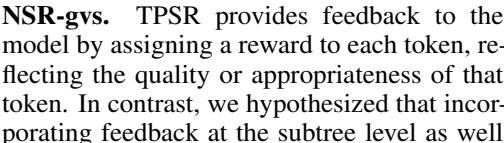

the following intuition: expressions that fit the same numerical data well are likely to share common substructures.

We first trained a slightly modified version of the NeSymReS model, where the model takes subtrees as prompts and generates expressions that incorporate them. We achieved this by extracting subtrees from the ground-truth expressions and feeding them to the model together with numerical data during training. Figure 3 illustrates the inference procedure of NSR-gvs. We generated multiple predictions iteratively by augmenting the model with varying prompts. For each iteration, we first sampled subtrees from a pool of candidate subtrees, which were extracted from high-performing expressions in previous predictions. To maintain output diversity, we also occasionally sampled subtrees from a random distribution. We then provided the sampled subtrees to the pre-trained model as prompts, along with the numerical data. After the model generates a prediction, it is automatically verified according to the fitting accuracy on the test data. Finally, the pool of candidate subtrees are updated based on the results of self-verification. This method can be formulated within the framework of reinforcement learning, and we provide a more detailed explanation in Appendix. B.

We conducted experiments in this section using 30 iteration loops per expression, with the beam size $b = 5$ for generating each prediction. In addition, we experimented with a method that combines NSR-gvs with TPSR; in this approach, each prediction is produced via MCTS-based decoding instead of simple beam search.

## 6.2 Results

We evaluated the impact of each test-time strategy on reproduction bias and numerical accuracy in a experimental setting. The experimental setup closely follows that described in Section 5.1. We trained a NeSym-ReS model and a prompt-augmented model for NSR-gvs with the same training dataset for the same number of epochs. We evaluated the strategies using the `not_included` dataset, where we used the $R^2$ metric to evaluate numerical accuracy, and the number of novel expressions to evaluate reproduction bias.

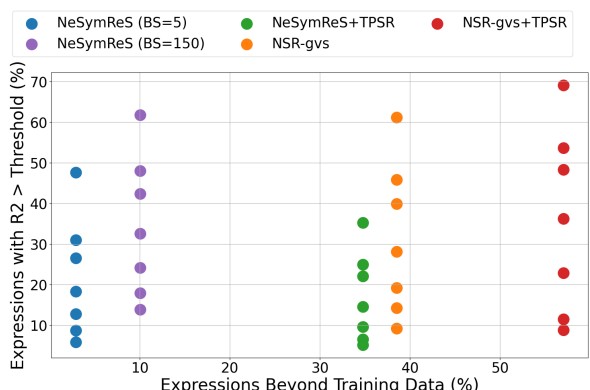

Figure 4 shows how the test-time strategies perform under the simplified setting. In terms of the ability to generate novel expressions, TPSR, NSR-gvs, and their combination demonstrate strong performance. These results imply that strategies involving the provision of additional information during inference (TPSR

Figure 4: Evaluation of test-time strategies on the `not_included` dataset. The x-axis represents the percentage of expressions generated that were not included in the training data. The y-axis shows the proportion of expressions that exceeded the $R^2$ thresholds of 0.5, 0.9, 0.95, 0.99, 0.999, 0.9999 and 0.99999, respectively.

and NSR-gvs) are more effective in reducing reproduction bias. However, the result shows that high novelty in generated expressions does not necessarily imply high numerical accuracy. In some cases, acquiring the ability to generate novel expressions leads to a decrease in numerical accuracy (TPSR), whereas some strategies can improve numerical accuracy despite high reproduction bias (large beam size).

## 6.3 Discussion

The experimental results show that while methods such as TPSR, which mitigate reproduction bias, can lead to a drop in numerical accuracy, decoding with large beam size improves numerical accuracy despite retaining a high level of reproduction bias. In this subsection, we discuss possible reasons why such phenomena occur.

Given that the additional information in TPSR and NSR-gvs is derived from self-verification, it should in theory offer better alternatives beyond the model's own logits, and is expected to assist in generating better expressions. Despite providing such useful information, the methods often under-perform in terms of numerical accuracy compared to the simple strategy of increasing beam size. This suggests that the Transformer struggles to leverage the additional information effectively, and in some cases, it might even be negatively impacted by it. For example, when the Transformer encounters an unfamiliar prefix within an partially constructed expression, it may become confused and could complete the expression with suboptimal tokens.

We therefore argue that providing additional information at test time in a way that is easy for the Transformer to leverage is important for developing a truly generalizable NSR approach. Viewed in this way, the use of subtrees at inference, as in the proposed method NSR-gvs, can be seen as a potentially valuable approach, since it contributes to mitigating reproduction bias and improving numerical accuracy.

## 7 CONCLUSION

In this work, we identified a major drawback of standard NSR models both theoretically and empirically. Our theoretical analysis shows that Transformers are incapable of generating expressions in a compositional way, while taking numerical data into account. We then examined the strategies that Transformers actually employ to generate expressions, and the results suggest that they mostly generate expressions copied from the training data, highly limiting the search space. We then demonstrate that incorporating additional information to the model during test-time can reduce reproduction bias. However, we also show that mitigating reproduction bias does not necessarily lead to higher numerical accuracy. The main limitation of this work is the absence of a method that simultaneously mitigates reproduction bias and improves numerical accuracy to a significant extent. In future work, we aim to build on the findings of this study to design symbolic regression methods with improved generalizability.

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

Table 2: Operators used in NeSymReS

| Arity | Operators |
|-------|-----------|
| Unary | pow2, pow3, pow4, pow5 |
|       | sqrt, log, exp |
|       | sin, cos, asin |
| Binary | add, sub, mul, div |

Table 3: Hyperparameters in NeSymReS's dataset generation

| Name | Explanation | Value |
|------|-------------|-------|
| $d$ | Dimension for input variables | 5 |
| $n$ | Number of input points | Sampled from $\mathcal{U}(1, 1000)$ |
| $\mathcal{D}_{\mathrm{mul}}$ | Distribution over multiplicative constants | Sampled from $\mathcal{LU}(0.05, 10)$ |
| $\mathcal{D}_{\mathrm{add}}$ | Distribution over additive constants | Sampled from $\mathcal{U}(-10, 10)$ |
| $\{x_{\min,j}\}_{j=1}^d$ | Lower bound for sampling input variable | Sampled from $\mathcal{U}(-10, 9)$ |
| $\{x_{\max,j}\}_{j=1}^d$ | Upper bound for sampling input variable | Sampled from $\mathcal{U}(x_{\min,j} + 1, 10)$ |

## A   DETAILS FOR NESYMRES

In this section, we present a detailed explanation for the study of NeSymReS that could not be fully explained in Section 3. We discuss the details of the dataset generation process, the model architecture, and the training procedure.

**Generating the dataset.**   In the first step for generating the expression $e^*$, the unary-binary tree structure $T$ is generated randomly within the limits of a maximum depth of 6. In the third step, the total number of constants added to the expression is also limited to a maximum of 6. The binary and unary operators $\mathcal{O}_{\mathrm{bin}} \cup \mathcal{O}_{\mathrm{un}}$ are shown in Table 2. Other hyparparameters are specified in Table 3, where $\mathcal{LU}$ denotes the log-uniform distribution.

**Model architecture.**   The NeSymReS model consists of two architectural components: the numerical encoder $enc_{\mathrm{num}}$ and a decoder $dec$. The numerical encoder processes the numerical data $\mathcal{D}$, represented as a tensor of shape $(b, n, d)$, where $b$ denotes the batch size, $n$ the number of input points, and $d$ the sum of dependent and independent variables. First, an embedding layer converts the numerical data into a higher dimensional tensor $\mathcal{D}'$ of shape $(b, n, h)$. This tensor is then processed by a 5-layer set-transformer (Lee et al., 2019) encoder that outputs a new tensor $Z_{num}$ of shape $(b, s, h)$, where $s$ denotes the number of embedding vectors produced by the encoder. The resulting tensor $Z_{num}$ is subsequently passed to the decoder $dec$, a five-layer standard Transformer decoder that auto-regressively generates the corresponding expression token by token. We set $b = 200$, $h = 512$, and $s = 32$ for our experiments.

**Details for training.**   During training, cross-entropy loss is used as the objective function, and teacher forcing (Sutskever et al., 2014) is applied during next-token prediction. The AdamW (Loshchilov & Hutter, 2017) optimizer is employed with an initial learning rate of $10^{-4}$. After 4000 steps, the learning rate is adjusted proportionally to the inverse square root of the number of steps taken.

## B   DETAILS FOR TEST-TIME STRATEGIES

This section is devoted to supplementing the details that were not fully covered in Section 6. We first supplement our explanation of TPSR, followed by a detailed formulation of NSR-gvs.

## B.1 TPSR

We detail the exact procedure for computing the reward in TPSR. As explained in Section 6, the reward is mainly calculated based on the generated expression's numerical accuracy, with additional consideration given to its complexity. In TPSR, a hyperparameter $\lambda$ controls the balance between fitting accuracy and complexity. Given a set of $n$ data points $\{\mathbf{x}_i, y_i\}_{i=1}^{n}$, and a candidate prediction $\tilde{f}$, the reward $r(\tilde{f}(\cdot) \mid \mathbf{x}, \mathbf{y})$ is calculated as follows:

$$r(\tilde{f}(\cdot) \mid \mathbf{x}, \mathbf{y}) = \frac{1}{1 + \mathrm{NMSE}(\mathbf{y}, \tilde{f}(\mathbf{x}))} + \lambda \exp\left(-\frac{|\mathrm{seq}(\tilde{f})|}{L}\right),$$

where seq is the serialization mapping introduced in Section 3, $L$ denotes the the model's maximum sequence length, and NMSE represents the normalized mean square loss. In our work, we always set $\lambda$ to 0.01, which is the default value in the original study of TPSR.

## B.2 NSR-GVS

As described in Section 6, NSR-gvs is a method that iteratively improves its predictions by providing expression subtrees as prompts to the model and receiving feedback through verification. In this section, we formulate the training and inference procedures of NSR-gvs within the framework of reinforcement learning.

### B.2.1 TRAINING

We first introduce a prompt-conditioned symbolic regressor $S'_{\boldsymbol{\theta}}$ defined by parameters $\boldsymbol{\theta}$, that maps a numerical dataset $\mathcal{D}$ and an auxiliary prompt sequence $\mathbf{p}$ to a symbolic expression $\hat{e} = S'_{\boldsymbol{\theta}}(\mathcal{D}, \mathbf{p})$. Learning aims to align $\hat{e}$ with the ground-truth expression $e^*$ underlying $\mathcal{D}$. Among the elements of the synthetic training tuple $(e^*, \mathcal{D}, \mathbf{p})$, the generation of $e^*$ and $\mathcal{D}$ is the same as explained in Section 3. Here we specify how prompt sequences are constructed.

We first define $\mathrm{extract} : \mathcal{E} \longrightarrow \mathcal{P}(\mathcal{E})$ as a stochastic mapping, which assigns to each symbolic expression $e \in \mathcal{E}$ a probability distribution over the subtrees of $e$. The space $\mathcal{P}(\mathcal{E})$ denotes the power set of expressions.

Using this stochastic mapping, we first obtain $N$ subtrees $\{e'_i \mid e'_i \sim \mathrm{extract}(e^*), i = 1, 2, \ldots, N\}$ from the ground-truth expression $e^*$. Then, each of the subtrees are converted to token sequences $\{\mathbf{t}_i \mid \mathbf{t}_i = \mathrm{seq}(e'_i), i = 1, 2, \ldots, N\}$ using the serialization map seq. Given an expression $e^*$, we construct the prompt:

$$\mathbf{p} = (\tau_{\mathrm{start}}, \mathbf{t}_1, \tau_{\mathrm{end}}, \tau_{\mathrm{start}}, \mathbf{t}_2, \tau_{\mathrm{end}}, \ldots, \tau_{\mathrm{start}}, \mathbf{t}_N, \tau_{\mathrm{end}}),$$

where tokens $\tau_{\mathrm{start}}$ and $\tau_{\mathrm{end}}$ are partition tokens representing the beginning and end of each subtree representation.

Similar to the formulation in Section 3, the predictive distribution $q'_{\boldsymbol{\theta}}(\cdot \mid (\mathbf{p}, \mathbf{s}_{<j}), \mathcal{D})$ is realized by an encoder-decoder Transformer parametrized by $\boldsymbol{\theta}$. In NSR-gvs, however, the decoder is conditioned on $(\mathbf{p}, \mathbf{s}_{<j})$, which is the concatenation of the prompt $\mathbf{p}$ and previously emitted prefix $\mathbf{s}_{<j}$.

### B.2.2 INFERENCE

During inference, we guide the symbolic regressor $S'_{\boldsymbol{\theta}}$ by prompting it with expression subtrees, which are obtained by a self-verification process. We formalize the inference-time mechanism of NSR-gvs within the framework of a Markov Decision Process (MDP). The core components of the MDP are defined as follows:

**State space $\mathcal{S}$ and action space $\mathcal{A}$.** The state at time $t$ is denoted by $s_t \in \mathcal{S}$. The state is defined as $s_t = \{(e'_i, z_i, c_i) \mid i = 1, 2, \ldots, n_t\}$, which is a $n_t$-sized set comprising tuples of subtrees $e'_i \in \mathcal{E}$, its corresponding verification scores $z_i \in \mathbb{R}$, and its appearance count $c_i \in \mathbb{N}$. Therefore, the state space can be represented as $\mathcal{S} = \mathcal{P}(\mathcal{E} \times \mathbb{R} \times \mathbb{N})$. The action $a_t \in \mathcal{A}$ is a prompt sequence described in the previous subsection. The action space is represented as $\mathcal{A} = (\Gamma \cup \{\tau_{\mathrm{start}}, \tau_{\mathrm{end}}\})^*$.

Table 4: Hyperparameters in NSR-gvs

| Name | Explanation | Value |
|---|---|---|
| $k$ | Size of high-scored subtree set $E'_{\text{topk}}$ | 39 |
| $k_{\text{rand}}$ | Size of randomly sampled subtree set $E'_{\text{rand}}$ | 9 |
| $z_{\text{thres}}$ | Threshold value for high-scored subtrees | 0.213 |
| $l_{\max}$ | Maximum length of a subtree's representation | 9 |
| $l_t$ | Total length of the subtrees' representation | Sampled from $\mathcal{U}(0, \lfloor 15.58 + 0.42t \rfloor)$ |

**Policy $\pi(a_t \mid s_t)$.** We define a stochastic policy to sample an action $a_t$ from the current state $s_t$. An action is sampled following the procedure below.

First, we deterministically select a set of subtrees $E'_{\text{topk}}$, consisting of the top $k$ subtrees with the highest verification scores in state $s_t$, as follows:

$$E'_{\text{topk}} = \{e'_i \mid (e'_i, z_i, c_i) \in s_{\text{topk}}, i = 1, 2, \ldots, k\}, \quad \text{where} \quad s_{\text{topk}} = \operatorname*{argmax}_{s \subseteq s_t, |s|=k} \sum_{(e',z,c) \in s} z.$$

Subsequently, we filter out subtrees whose corresponding score $z$ is smaller than a threshold value $z_{\text{thres}}$. The purpose of this operation is to prioritize exploration over exploitation when the quality of obtained subtrees are poor.

Next, we construct a set $E'_{\text{rand}}$ by extracting $k_{\text{rand}}$ subtrees from expressions sampled from the expression generator $p_{\mathcal{E}}$:

$$E'_{\text{rand}} = \{e'_i \mid e'_i \sim \text{extract}(e), e \sim p_{\mathcal{E}}, i = 1, 2, \ldots, k_{\text{rand}}\}.$$

Finally, we uniformly sample a set of subtrees from the merged set $E'_{\text{topk}} \cup E'_{\text{rand}}$ and convert them to tokens in the same way as during training time, resulting in a prompt sequence $a_t$. During sampling, we filter out subtrees whose token representation is longer than $l_{\max}$, and we sample subtrees until the total length of the subtrees' token representation exceeds the limit $l_t$.

By sampling from both the self-verification-based set $E'_{\text{topk}}$ and the randomly obtained set $E'_{\text{rand}}$, the policy enables both exploration and exploitation. The hyperparameters $k$, $k_{\text{rand}}$, $z_{\text{thres}}$, $l_{\max}$, and $l_t$ characterize the policy.

**Reward function $R(a_t, s_t)$ and transition probability $T(s_{t+1} \mid a_t, s_t)$.** After an action $a_t$ is sampled, it is provided to the prompt-conditioned symbolic regressor $S'_{\boldsymbol{\theta}}$ together with numerical data $\mathcal{D}$. We compute the reward based on the numerical accuracy of the prediction $\hat{e} = S'_{\boldsymbol{\theta}}(\mathcal{D}, a_t)$.

Let $\mathcal{L} : \mathbb{R}^n \times \mathbb{R}^n \to \mathbb{R}$ be a metric to evaluate the difference between two vectors (in practice, we use the $R^2$ value described in Section 5). When $\mathcal{D} = \{(\mathbf{x}_i, y_i)\}_{i=1}^n$, the reward is computed as:

$$R(a_t, s_t) = \mathcal{L}(\mathbf{y}, \hat{e}(\mathbf{x})).$$

Finally, we define the transition probability $T(s_{t+1} \mid a_t, s_t)$, determined by the following process. We denote by $\hat{E}'$ the set comprising all subtree expressions of $\hat{e}$. For each subtree $\hat{e}'$ in $\hat{E}'$, we update $s_t$ so that the verification score of each subtree matches the average reward of all expressions that included the subtree, as described below.

1. If $\forall (e', z, c) \in s_t, \hat{e}' \neq e'$ holds, add the tuple $(\hat{e}', R(a_t, s_t), 1)$ to $s_t$.

2. If $\exists (e', z, c) \in s_t, \hat{e}' = e'$ holds, replace the tuple $(\hat{e}', z, c)$ with $(\hat{e}', \dfrac{cz + R(a_t, s_t)}{c + 1}, c + 1)$.

The updated state serves as the state $s_{t+1}$ at the next timestep $t + 1$.

The overall algorithm during inference-time is detailed in 1. For the hyperparameters that characterize the policy, we use the values shown in Table 4, which were tuned via Bayesian optimization on 5 randomly generated expressions. The function $\lfloor \cdot \rfloor$ indicates the floor function, which rounds down the input to its nearest integer.

---

**Algorithm 1** Inference-time Algorithm

---

**function** VERIFY$(e, \mathcal{D})$
    $(X, \mathbf{y}) \leftarrow \mathcal{D}$
    $\hat{\mathbf{y}} \leftarrow e(X)$
    Compute $R^2$ score between $\mathbf{y}$ and $\hat{\mathbf{y}}$
    **return** $R^2$

**function** UPDATE$(e, s_t, R^2)$
    $E'_p \leftarrow$ Partial expressions extracted from $e$
    $s_{t+1} \leftarrow [\,]$
    **for** $(e_p, z, c)$ in $s_t$ **do**
        **if** $e_p \in E'_p$ **then**
$$z \leftarrow \frac{cz + R^2}{c+1}$$
            $c \leftarrow c + 1$
        Append $(e_p, z, c)$ to $s_{t+1}$
    **return** $s_{t+1}$

**procedure** NSR-GVS-INFERENCE$(\mathcal{D})$
    $s_1 \leftarrow [\,]$
    $e_{\text{best}} \leftarrow$ None
    $R^2_{\text{best}} \leftarrow -\infty$
    **for** $t \leftarrow 1$ to $T$ **do**
        **if** $t = 1$ **then**
            $a_t \leftarrow [\,]$
        **else**
            $E'_{\text{topk}} \leftarrow$ Top $k$ expressions in $s_t$ with high score
            Filter out expressions in $E'_{\text{topk}}$ whose corresponding score $z < z_{\text{thres}}$
            $E'_{\text{rand}} \leftarrow$ Randomly sampled partial expressions
            $E'_{\text{merged}} \leftarrow E'_{\text{topk}} \cup E'_{\text{rand}}$
            $a_t \leftarrow$ Uniformly sampled subset from $E'_{\text{merged}}$, converted to tokens
        $\mathbf{s} \leftarrow$ Transformer$(\mathcal{D}, a_t)$
        Convert sequence $\mathbf{s}$ to expression $e$
        $R^2 \leftarrow$ Verify$(e, \mathcal{D})$
        $s_{t+1} \leftarrow$ Update$(e, s_t, R^2)$
        **if** $R^2 > R^2_{\text{best}}$ **then**
            $e_{\text{best}} \leftarrow e$
            $R^2_{\text{best}} \leftarrow R^2$
    **return** $e_{\text{best}}$

---

## C  DETAILS FOR EXPERIMENTS, IMPLEMENTATION, AND USE OF LLMs

In this section, we describe the details for the experiments conducted in Section 5, 6, and D. We also provide details regarding our implementation and the computational resources used in our experiments.

We provide the model with 100 data points in all experiments. We selected the range of the data support as follows: for the AI Feynman dataset, we used the support defined by the dataset itself. For all other datasets, we sampled the support range using the same procedure as used when generating the training data. For error bars, we report the standard deviation across three different random seeds. For the method combining NSR-gvs and TPSR, however, we conducted experiments with only a single seed due to the long inference time. Our implementation for data generation, model training, and related components is based on the original implementation of NSRwH [1]. For the transformer4sr and TPSR experiments, we used the official implementation provided by the authors [2], [3]. For both implementations, we used the version of the implementation that was available on May 15. 2025. We trained and tested the model on a single NVIDIA A100 GPU. Training requires approximately 24 hours either for 1000 epochs on a dataset with $100,000$ expressions or for 10 epochs on a dataset with 10 million expressions. The time required to generate a single expression at test time is less than one minute when using only NeSymReS or NSRwH, approximately 3 to 10 minutes with TPSR or NSR-gvs, and around 2 to 5 hours when combining TPSR with NSR-gvs.

We used large language models (LLMs) to aid writing and coding, where we mainly used Gemini 2.5 Flash and GPT-5 to generate code and check on errors in writing.

## D  ADDITIONAL EXPERIMENTS

### D.1  VARYING THE TRAINING DATASET SIZE IN TRANSFORMER4SR

In Section 5, we tested whether reproduction bias occurs in the practical setting of transformer4sr. Although the dataset size that we tested on was fairly large (1.5M expressions), there is a possibility that further scaling the dataset size alone can mitigate reproduction bias. We therefore construct multiple training datasets with varying size to examine how reproduction bias trends as the dataset size increases. We construct datasets with the size ranging from 100K to 1.5M and present the result in Figure 5. The result shows that increasing the training dataset size does mitigate reproduction bias at the start, but not necessarily after a certain limit to the training dataset size.

### D.2  NUMERICAL ACCURACY IN PRACTICAL SETTINGS

We additionally evaluate and compare the numerical performance of the test-time strategies under conditions that better reflect practical applications. A total of 10 million expressions were used to construct the training dataset, employing all operators described in Section A without any restriction on operator types. We trained both a NeSymReS model and a prompt-augmented model on this dataset for 10 epochs. For the test datasets, we prepared the following two sets:

- `AI-Feynman`. This dataset consists of 91 equations with up to five independent variables, extracted from the AIFeynman database (Udrescu & Tegmark, 2020). It is commonly used in various studies to assess the performance of symbolic regression methods.

- `only_five_variables_nc`. This dataset consists of expressions containing exactly five independent variables, making it a challenging dataset. The "nc" designation indicates that the expressions do not include constants, which simplifies the problems slightly; however, it remains more difficult than the first dataset. The dataset was constructed by sampling expressions from $p_\mathcal{E}$, filtering for expressions that include exactly five variables, and finally deleting its constants. This dataset is derived from the study of NSRwH (Bendinelli et al., 2023), and we use the first 100 expressions for evaluation.

---

[1]https://github.com/SymposiumOrganization/ControllableNeuralSymbolicRegression
[2]https://github.com/omron-sinicx/transformer4sr
[3]https://github.com/deep-symbolic-mathematics/TPSR

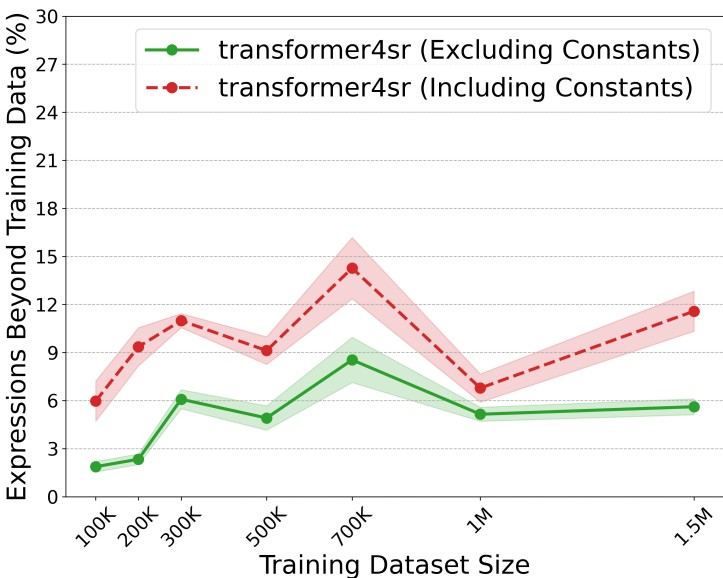

Figure 5: Reproduction bias in transformer4sr with varying training dataset sizes. While small training dataset sizes (100K, 200K) exhibit stronger reproduction bias, scaling the training dataset size does not necessarily mitigate reproduction bias after a certain limit.

- `black-box`. We also evaluated on numerical data collected from the real world, whose ground-truth expressions do not exist. We extracted 35 expressions from the black-box dataset in SR-Bench (La Cava et al., 2021) whose number of independent variables are five or less. The data are often noisy and may be sampled from a range different from the numerical data that the models were trained on, making the task challenging for the test-time computation methods.

Figure 6 demonstrates how the different test-time strategies perform under more practical settings. TPSR relatively performs slightly better than in the controlled setting; however, the general pattern of numerical accuracy remains consistent. These results demonstrate that, even in practical settings, test-time strategies that mitigate reproduction bias do not always result in better performance.

The result for the `black-box` dataset is shown in Figure 7. Consistent with the results above, NSR-gvs improves performance, and combining it with TPSR leads to further gains. This result show how NSR-gvs can improve performance robustly even on noisy datasets with the range of numerical data different from training time. TPSR also improves performance to a certain extent in this case.

### D.3 TRADE-OFF BETWEEN PERFORMANCE AND COMPUTATIONAL COST

The results in Section 6 show how the relationship between reproduction bias and numerical accuracy differ between various test-time strategies. However, test-time strategies also differ in terms of the computational cost required to generate an expression. In this section, we aim to better understand each test-time strategy by analyzing the trade-off between performance and the computational cost of expression generation. We also varied the beam size during decoding for NeSymReS, TPSR, and NSR-gvs for a more comprehensive analysis. We tested under the controlled setting described in Section 5, using the `not_included` dataset as the test dataset.

To measure the computational cost, we followed the approach of Shojaee et al. (2023) and used the number of candidate expressions generated by the model during the generation of a single equation. For example, this value corresponds to the beam size in NeSymReS, the number of total rollouts multiplied by beam size in TPSR, and the number of iteration loops multiplied by beam size in NSR-gvs.

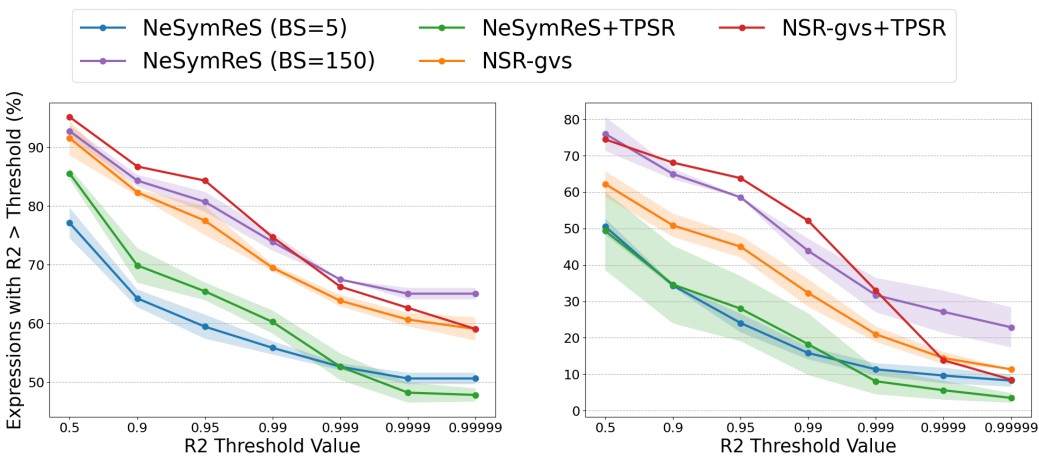

Figure 6: Comparison of test-time strategies under practical settings. The figure on the left shows the performance on the `AI-Feynman` dataset, and the figure on the right presents results on the `only_five_variables_nc` dataset.

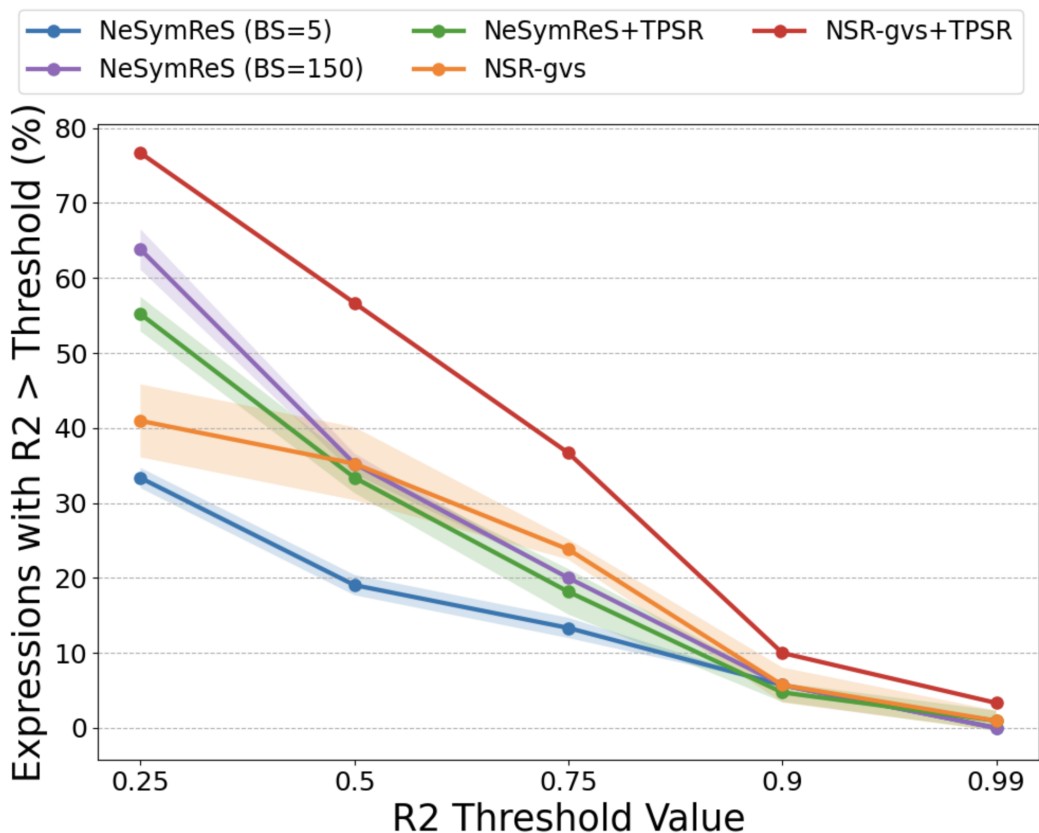

Figure 7: Comparison of test-time strategies under practical settings. The figure shows the performance on the 35 expressions with less than five independent variables extracted from the black-box dataset in SR-Bench.

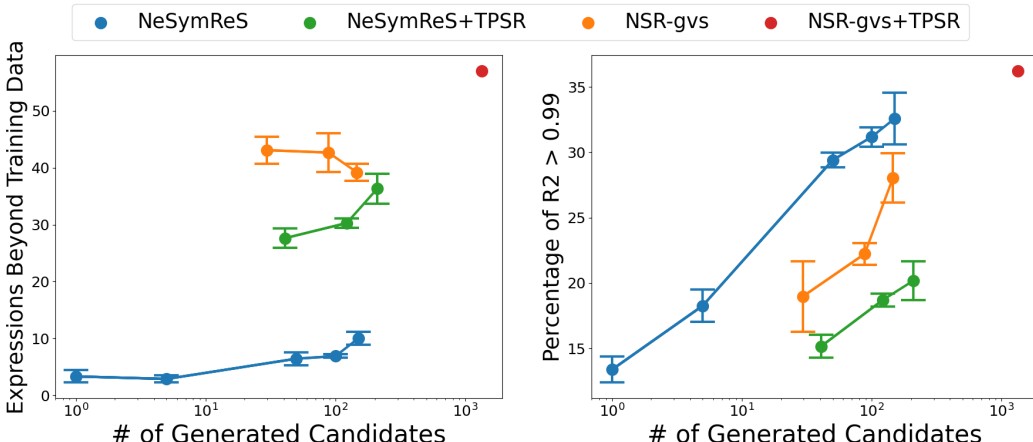

Figure 8: Trade-off between performance and computational cost for different test-time strategies. We varied the beam sizes for each model as follows: $\{1, 5, 50, 100, 150\}$ for NeSymReS, and $\{1, 3, 5\}$ for both NeSymReS+TPSR and NSR-gvs. For NSR-gvs+TPSR, we only experimented with beam size set to $1$. The left figure shows the trade-off between the ability to generate expressions and computational cost, while the right figure shows the trade-off between numerical accuracy and computational cost.

We present the results in Figure 8. It can be observed that, unlike NeSymReS—where larger beam size yields only limited reduction in reproduction bias—TPSR and NSR-gvs achieve notable reductions in reproduction bias at comparable computational costs. However, in terms of numerical accuracy, simply increasing the beam size in NeSymReS yields better performance than using NSR-gvs or TPSR at a comparable computational cost. The results support the conclusion in Section 6 that the reduction of reproduction bias is only weakly correlated with numerical accuracy.

### D.4 CAN NSRwH ALSO MITIGATE REPRODUCTION BIAS?

When researchers in fields of natural sciences or engineering model their experimental data, they often make use of prior knowledge. For example, scientists may anticipate a symmetry between variables or predict that a particular operator appears in the mathematical laws describing the data. NSRwH (Bendinelli et al., 2023) is a method that enables incorporating such prior knowledge into the NeSymReS model. The types of prior knowledge provided to the model include the following:

- **Complexity.** The complexity of an expression is defined by the number of tokens used in the expression's token sequence. The model is provided with the complexity of the ground-truth expression.

- **Symmetry.** The presence or absence of symmetry among the input variables is provided to the model.

- **Positives.** Subtrees appearing in the ground-truth expression are provided to the model. Additionally, the value of constants appearing in the ground-truth expression may also be provided.

- **Negatives.** Subtrees that do not appear in the ground-truth expression are provided to the model.

In NSRwH, prior knowledge is encoded by an additional symbolical encoder $enc_{\text{sym}}$. The output of the symbolical encoder is summed together with the output of NeSymReS's numerical encoder and is fed to the decoder.

While prior knowledge is required beforehand to use NSRwH, it is a method that provides the model with additional information during inference, similar to TPSR and NSR-gvs. In this section, we test

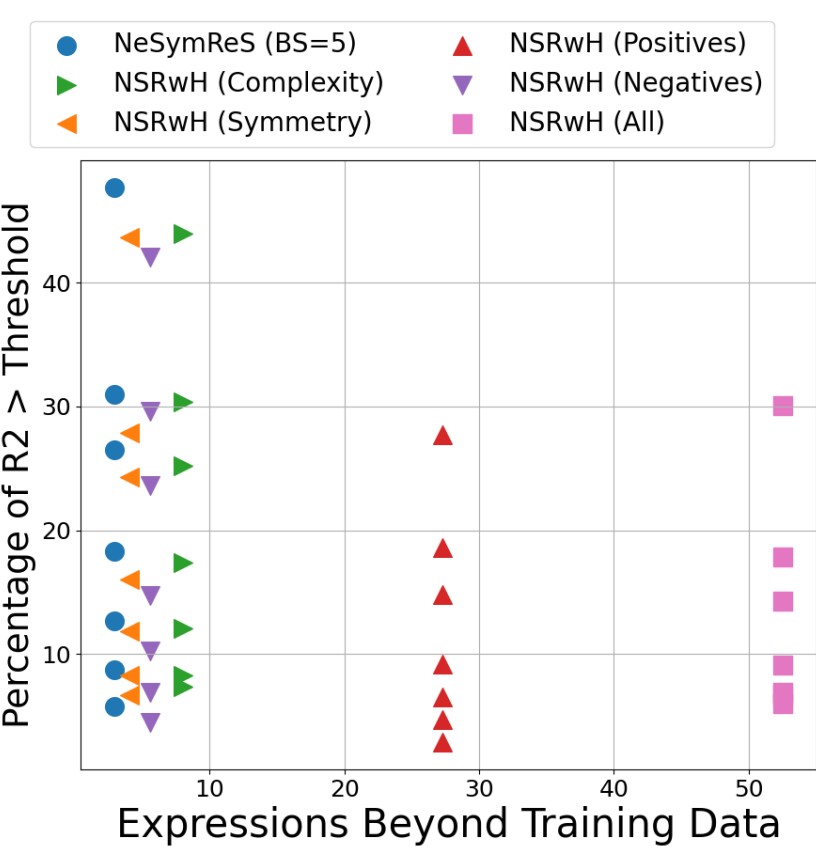

Figure 9: Evaluation of NSRwH on the `not_included` dataset. The x-axis represents the percentage of expressions generated that were not included in the training data. The y-axis shows the proportion of expressions that exceeded the $R^2$ thresholds of 0.5, 0.9, 0.95, 0.99, 0.999, 0.9999 and 0.99999, respectively.

Table 5: Breakdown of generated expressions by novelty and high accuracy ($R^2 > 0.99$) across test-time strategies

| Test-time Strategy | Novel, $R^2 > 0.99$ | Novel, $R^2 \leq 0.99$ | Not Novel, $R^2 > 0.99$ | Not Novel, $R^2 \leq 0.99$ |
|---|---|---|---|---|
| NeSymReS (BS=1) | 0.45 | 2.89 | 12.92 | 83.74 |
| NeSymReS (BS=5) | 0.67 | 2.23 | 17.59 | 79.52 |
| NeSymReS (BS=50) | 4.23 | 2.23 | 25.17 | 68.38 |
| NeSymReS (BS=100) | 4.45 | 2.45 | 26.72 | 66.37 |
| NeSymReS (BS=150) | 6.25 | 3.79 | 26.33 | 63.62 |
| **NeSymReS + TPSR (BS=1)** | 0.45 | **27.17** | 14.70 | 57.69 |
| **NeSymReS + TPSR (BS=3)** | 0.67 | **29.62** | 18.04 | 51.67 |
| **NeSymReS + TPSR (BS=5)** | 2.02 | **34.31** | 18.16 | 45.51 |
| **NSR-gvs (BS=1)** | 4.91 | **38.18** | 14.05 | 42.86 |
| **NSR-gvs (BS=3)** | 6.67 | **36.00** | 15.56 | 41.78 |
| **NSR-gvs (BS=5)** | 8.68 | **30.51** | 19.38 | 41.43 |
| **NSR-gvs + TPSR (BS=1)** | 12.75 | **44.30** | 23.46 | 19.46 |
| NSRwH (Complexity, BS=5) | 2.23 | 5.80 | 15.18 | 76.78 |
| NSRwH (Symmetry, BS=5) | 1.11 | 2.90 | 14.93 | 81.06 |
| **NSRwH (Positives, BS=5)** | 2.46 | **24.83** | 6.71 | 66.00 |
| NSRwH (Negatives, BS=5) | 0.89 | 4.68 | 13.81 | 80.62 |
| **NSRwH (All, BS=5)** | 4.90 | **47.66** | 4.23 | 43.21 |

whether NSRwH can mitigate reproduction bias when prior knowledge is provided. We obtained a NSRwH model by finetuning the NeSymReS model that we trained in Section 5. We froze the numerical encoder of the NeSymReS model, attached a symbolical encoder, and finetuned the model for 250 epochs. We used the same training dataset as in Section 5 consisting of $100,000$ expressions; however, during fine-tuning, prior knowledge was extracted from the ground-truth expressions and fed into the symbolic encoder. At test time, we evaluated the NSRwH model under settings where each type of prior knowledge is provided individually, as well as under a setting where all types of prior knowledge are provided simultaneously. We follow the default settings of NSRwH to determine the amount of prior knowledge provided during test-time, and we used the `not_included` dataset as the test dataset. We set the beam size to 5 and compare the results with those of NeSymReS, which is also configured with a beam size of 5.

Figure 9 shows the results for this experiment. While providing complexity, symmetry, or absent subtrees mitigates reproduction bias only to a limited extent, providing appearing subtrees or providing all properties significantly mitigates reproduction bias. However, we also observe that the numerical accuracy of NSRwH decreases when provided with appearing subtrees or with all properties. This indicates a limitation of NSRwH when dealing with data not included in the training set. The results also show that not all kinds of additional data are effective for mitigating reproduction bias.

### D.5 Do Novel Expressions Contribute to Improvements in Numerical Accuracy?

In Section 6, we saw that providing additional information to the model during inference can lead to generation of novel expressions. However, we also demonstrated that mitigating reproduction bias does not necessarily lead to better numerical accuracy. In this section, we analyzed how much the novel expressions generated under each test-time strategy (including NSRwH) contribute to improvements in numerical accuracy, and present the corresponding results in Table 5. "Novel" indicates that the generated expression does not appear in the training data, while "Not Novel" means it does. The values indicate the percentage of expressions that satisfy each condition.

The results show that for test-time strategies that are capable of mitigating reproduction bias (strategies shown in bold), a large proportion of generated novel expressions do not perform well in terms of numerical accuracy. Especially for TPSR, hardly any of the novel expressions exhibit high numerical accuracy. This indicates the difficulty of generating appropriate expressions from an expanded search

space. However, for strategies using NSR-gvs, novel expressions contribute to high accuracy to some extent, showing that additional information can be beneficial for both mitigating reproduction bias and improving numerical accuracy in some occasions.

### D.6 FURTHER RESULTS ON THE `BASELINE` DATASET

As described in Section 5, the empirical results show that the `baseline` dataset is a much more easier dataset compared to the `not_included` dataset with naive inference. In this section, we present the results concerning the numerical accuracy for various test-time strategies on the `baseline` dataset. We also test with NSRwH as well as the test-time strategies described in Section 6.

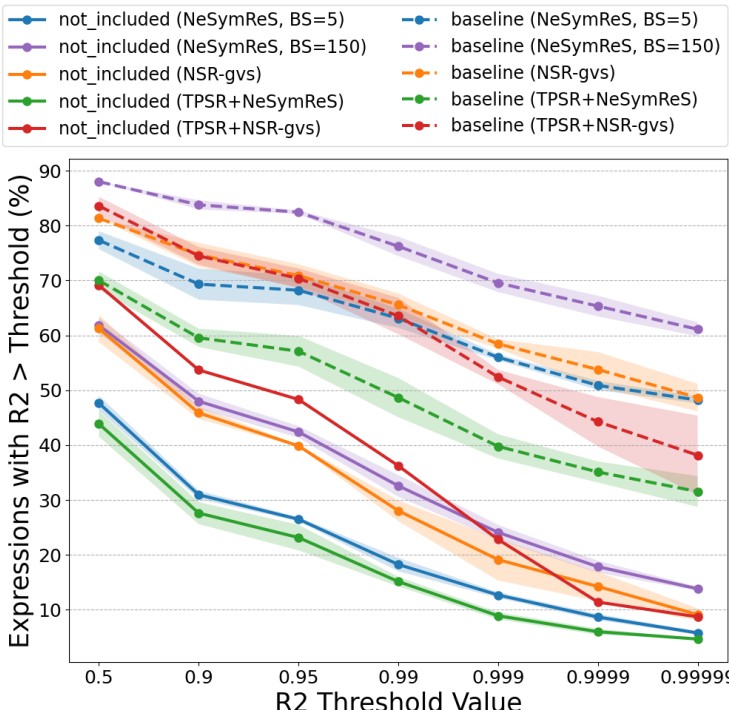

Figure 10: The y-axis shows the proportion of expressions that exceeded the $R^2$ thresholds of 0.5, 0.9, 0.95, 0.99, 0.999, 0.9999 and 0.99999, respectively.

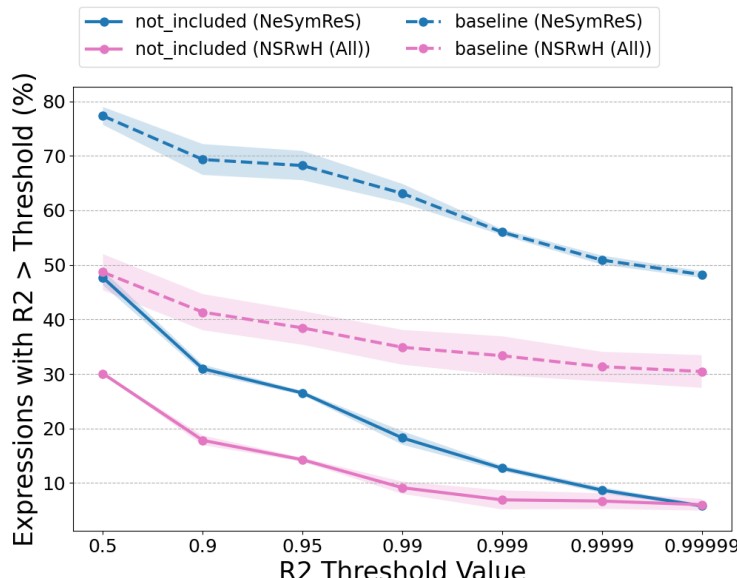

Figure 11: The y-axis shows the proportion of expressions that exceeded the $R^2$ thresholds of 0.5, 0.9, 0.95, 0.99, 0.999, 0.9999 and 0.99999, respectively.

# E    THEORETICAL BACKGROUND AND PROOF

In this section, we provide background knowledge, detailed settings, and a complete proof for the theoretical result presented in Section 5.

## E.1    PRELIMINARY

We first provide a brief overview of relevant circuit complexity classes. We then define the class of log-precision Transformers and introduce its simulation guarantees. We also present a formal definition of the Boolean formula value problem, which we use in our proof.

### E.1.1    CIRCUIT COMPLEXITY CLASSES

We offer an explanation to several fundamental circuit complexity classes that are used in our theoretical analysis. Particularly, we discuss the complexity classes $\mathsf{AC}^0, \mathsf{TC}^0$ and , $\mathsf{NC}^1$. The relationship between these three classes can be summarized as follows:

$$\mathsf{AC}^0 \subsetneq \mathsf{TC}^0 \subset \mathsf{NC}^1.$$

Whether $\mathsf{TC}^0$ is a proper subset of $\mathsf{NC}^1$ is an open question, but it is widely believed that this is the case. For a more detailed and comprehensive introduction, we recommend reference to (Arora & Barak, 2009).

**Circuit class $\mathsf{AC}^0$ .**    The class $\mathsf{AC}^0$ consists of Boolean circuits of constant depth and polynomial size whose gates have unbounded fan-in and are restricted to the basis $\overline{\{\mathsf{AND}, \mathsf{OR}, \mathsf{NOT}\}}$. Intuitively, $\mathsf{AC}^0$ captures extremely shallow parallel computation.

**Circuit class $\mathsf{TC}^0$ .**    The class $\mathsf{TC}^0$ is an extension of $\mathsf{AC}^0$, where a gate called the majority gate can be additionally used. A majority gate has unbounded fan-in and outputs false when half or more of the inputs are false, and true otherwise. Other definitions are the same as $\mathsf{AC}^0$.

**Circuit class $\mathsf{NC}^1$ .**   Circuits in $\mathsf{NC}^1$ are polynomial sized with the depth logarithmic to the input size. They comprise of $\{\mathsf{AND}, \mathsf{OR}, \mathsf{NOT}\}$ gates with constant fan-in. The class $\mathsf{NC}^1$ contains several well-known problems such as the parity check on a bit string.

### E.1.2   Log-precision Transformers

We assume bounded-depth log-precision Transformers throughout the theoretical analysis. We first model the parametrized Transformer $\mathsf{TF}_{\boldsymbol{\theta}}$ as a next-token prediction function;

$$\mathsf{TF}_{\boldsymbol{\theta}} : \Gamma^m \times \mathbb{R}^{(d+1) \times n} \longrightarrow \Gamma, \tag{2}$$

i.e. the Transformer receives a length-$m$ prefix along with a numerical dataset $\mathcal{D}$ and outputs a single token $u \in \Gamma$.

**Definition 2** $((D, d)$-bounded log-precision Transformer). *Let $k$ be the input length. A $(D, d)$-bounded log-precision Transformer is an encoder–decoder model that satisfies*

1. *constant depth $D = O(1)$,*

2. *hidden size $d \leq Q(k)$ for a fixed polynomial $Q$,*

3. *the values at all layers, as well as the outputs of all key intermediate operations in it (attention, activation, arithmetic operators, etc.), are represented using $O(\log k)$ bits.*

For specific definitions of operations that enable approximation in $O(\log k)$ bits, please refer to Section 4 and Appendix A of Merrill & Sabharwal (2023b). We introduce the simulation guarantees for bounded-depth log-precision Transformers as follows.

**Lemma 1** (Circuit simulation (Merrill & Sabharwal, 2023b, Cor. 2.1)). *Any $(D, d)$-bounded log-precision Transformer can be simulated by a family of $\mathsf{TC}^0$ circuits of size $\mathrm{poly}(k)$ and constant depth with respect to $k$.*

### E.1.3   The Boolean Formula Value Problem

Following the definition by Buss (1987), we introduce the definition of the Boolean formula value problem as follows.

**Definition 3** (Boolean formula value problem). *Let $\Lambda = \{0, 1, \wedge, \vee, \neg, (, )\}$ be the alphabet. A Boolean formula is a string defined recursively as follows:*

1. *0 and 1 are Boolean formulae;*

2. *If $\mathbf{t}_1$ and $\mathbf{t}_2$ are two Boolean formulae, then $(\neg \mathbf{t}_1), (\mathbf{t}_1 \wedge \mathbf{t}_2), (\mathbf{t}_1 \vee \mathbf{t}_2)$ are also Boolean formulae.*

*When given a boolean formula $\mathbf{t}$, the goal of the Boolean formula value problem is to compute whether the evaluation result $\mathrm{eval}(\mathbf{t})$ of a given Boolean formula is 0 or 1.*

### E.2   Main Theorem

Prior to proving the main theorem, we state the following Lemma from Feng et al. (2023). The detailed proof of this Lemma can be found in the same paper. The Lemma states that $\mathsf{TC}^0$ circuits are capable of identifying the indexes of paired brackets in a string.

**Lemma 2** (Bracket parsing (Feng et al., 2023, Lem. D.3)). *Consider any string $\mathbf{t} = t_1 t_2 \cdots t_n$ of length $n$ containing brackets '(', ')', and other characters, and all brackets in $\mathbf{t}$ are paired. Let $\boldsymbol{g}$ be a boolean function taking $\mathbf{t}$ as input and output $n$ pairs of integers defined as follows:*

$$\boldsymbol{g}_i(\mathbf{t}) = \begin{cases} (-1, j) & \text{if } t_i \text{ is a left bracket and } t_i, t_j \text{ are paired.} \\ (j, -1) & \text{if } t_i \text{ is a right bracket and } t_i, t_j \text{ are paired.} \\ (j, k) & \text{if } t_i \text{ is not a bracket, and } t_j, t_k \text{ are the nearest paired brackets containing } t_i. \end{cases}$$

*Then $\boldsymbol{g}$ can be implemented by the $\mathsf{TC}^0$ circuits.*

We now proceed to prove the main theorem of our theoretical analysis.

**Theorem 2** (Bounded log-precision Transformer lower bound). *Assume* $\mathsf{TC}^0 \neq \mathsf{NC}^1$. *For any integer $D$ and any polynomial $Q$, there exists a problem size $m$ such that no $(D, d)$-bounded log-precision Transformer with $d \leq Q(m)$ can solve* $\mathrm{LastTokenPrediction}(m)$.

*Proof.* Fix $D$ and $Q$ and suppose, for contradiction, that for some sufficiently large $m$ there exists an $(D, d)$-bounded log-precision Transformer $\mathsf{TF}_\theta$ with $d \leq Q(m)$ that solves the problem of $\mathrm{LastTokenPrediction}(m)$.

**Step 1 (simulation).** By Lemma 1, $\mathsf{TF}_\theta$ can be simulated by a $\mathsf{TC}^0$ circuit family of size $\mathrm{poly}(m)$. Hence, under our assumption, $\mathrm{LastTokenPrediction}(m) \in \mathsf{TC}^0$.

**Step 2 ($\mathsf{TC}^0$ construction).** We show that there exists a $\mathsf{TC}^0$ circuit that can translate any instance of a Boolean formula value problem to an instance of the last-token prediction problem.

Let $\mathbf{t}$ be a boolean formula. There exists a $\mathsf{TC}^0$ circuit that performs:

1. Translation of $\mathbf{t}$ to a $\vee$-free Boolean formula $\mathbf{t}'$.

2. Conversion of $\mathbf{t}'$ to its prefix notation $\mathbf{t}'_{\mathrm{pre}}$.

3. Conversion of $\mathbf{t}'_{\mathrm{pre}}$ to a token sequence $\mathbf{s} \in \Gamma^*$ by the following procedure:

    (a) replace 0 with $\{\times, x_1, x_2\}$ and 1 with $\{-, x_1, x_2\}$;
    (b) replace $\wedge$ with $\times$;
    (c) replace $\neg$ with $\{-, -, x_1, x_2\}$.

4. Local edits:

    (a) prepend $+$ to $\mathbf{s}$ to form the incomplete token sequence $\widetilde{\mathbf{s}}$;
    (b) set $n = 2$, and attach the data points $(x_{1,1}, x_{2,1}, y_1) = (1, 0, 1)$ and $(x_{1,2}, x_{2,2}, y_2) = (0, -1, 0)$ to the input numerical data $\mathcal{D}$;
    (c) define the metric as the mean squared error: $\mathcal{L}(\mathbf{y}, \hat{\mathbf{y}}) = \frac{1}{n} \sum_{i=1}^n (y_i - \hat{y}_i)^2$

To perform the first step, for all $\vee$ in $\mathbf{t}$, we must replace the nearest left bracket containing $\vee$ with $\neg(\neg$ and also replace $\vee$ with $\wedge\neg$. By using the results of Lemma 2, it follows that this operation can be performed by a circuit within $\mathsf{TC}^0$ complexity. The second step can be implemented by $\mathsf{AC}^0$ circuits, according to Buss (1987, Cor. 11). Since the third and fourth steps only involve replacing and extending obtained sequences, these steps can also be implemented by $\mathsf{AC}^0$ circuits.

**Step 3 (soundness of the reduction).** When $\mathrm{eval}(\mathbf{t}) = 0$, the losses $\mathcal{L}(\mathbf{y}, e_{(\widetilde{\mathbf{s}}, u)}(\mathbf{x}))$ for each leaf token $u \in \{x_1, x_2, C\}$ can be computed as follows:

$$
\begin{cases}
\mathcal{L}(\mathbf{y}, e_{(\widetilde{\mathbf{s}}, x_1)}(\mathbf{x})) = \dfrac{1}{2} \sum_{i=1}^2 (y_i - x_{1,i})^2 = \dfrac{1}{2}(0^2 + 0^2) = 0, \\[2ex]
\mathcal{L}(\mathbf{y}, e_{(\widetilde{\mathbf{s}}, x_2)}(\mathbf{x})) = \dfrac{1}{2} \sum_{i=1}^2 (y_i - x_{2,i})^2 = \dfrac{1}{2}(1^2 + 1^2) = 1, \\[2ex]
\mathcal{L}(\mathbf{y}, e_{(\widetilde{\mathbf{s}}, C)}(\mathbf{x})) = \underset{c \in \mathcal{C}}{\mathrm{argmin}}\ \dfrac{1}{2} \sum_{i=1}^2 (y_i - c)^2 = \underset{c \in \mathcal{C}}{\mathrm{argmin}}\ \dfrac{1}{2}((1 - c)^2 + (-c)^2) \geq \dfrac{1}{4},
\end{cases}
$$

where $\mathcal{C}$ is the interval from which numeric constants are drawn, and $e_{\mathbf{s}} = \mathrm{expr}(\mathbf{s}, \mathcal{D})$ is the mapping function defined in Section 5. When $\mathrm{eval}(\mathbf{t}) = 1$, the losses $\mathcal{L}(\mathbf{y}, e_{(\widetilde{\mathbf{s}}, u)}(\mathbf{x}))$ can be computed as

follows:

$$
\begin{cases}
\mathcal{L}(\mathbf{y}, e_{(\widetilde{\mathbf{s}}, x_1)}(\mathbf{x})) = \dfrac{1}{2} \sum_{i=1}^{2} (y_i - (1 + x_{1,i}))^2 = \dfrac{1}{2}((-1)^2 + (-1)^2) = 1, \\[3mm]
\mathcal{L}(\mathbf{y}, e_{(\widetilde{\mathbf{s}}, x_2)}(\mathbf{x})) = \dfrac{1}{2} \sum_{i=1}^{2} (y_i - (1 + x_{2,i}))^2 = \dfrac{1}{2}(0^2 + 0^2) = 0, \\[3mm]
\mathcal{L}(\mathbf{y}, e_{(\widetilde{\mathbf{s}}, C)}(\mathbf{x})) = \operatorname*{argmin}_{c \in \mathcal{C}} \dfrac{1}{2} \sum_{i=1}^{2} (y_i - (1 + c))^2 = \operatorname*{argmin}_{c \in \mathcal{C}} \dfrac{1}{2}((-c)^2 + (-1 - c)^2) \geq \dfrac{1}{4},
\end{cases}
$$

Consequently, when $\mathrm{eval}(\mathbf{t}) = 0$, the result for the corresponding last-token prediction problem is $u^* = x_1$, while when $\mathrm{eval}(\mathbf{t}) = 1$, the result is $u^* = x_2$. Hence the mapping introduced in Step 2 is a valid $\mathsf{TC}^0$ many-one reduction from the Boolean formula value problem to the last-token prediction problem.

**Step 4 (contradiction).** The Boolean formula value problem is $\mathsf{NC}^1$-complete under $\mathsf{AC}^0$ reductions (Buss, 1987, Thm. 9). Hence Step 2 and Step 3 indicate that $\mathrm{LastTokenPrediction}(m) \notin \mathsf{TC}^0$, contradicting Step 1 and the assumed strict inclusion $\mathsf{TC}^0 \subsetneq \mathsf{NC}^1$. Therefore, such a Transformer cannot exist. □

### E.3 PAC APPROXIMATION VIA ITERATED SELF-VERIFICATION

We further present theoretical analysis regarding the performance of the proposed method, NSR-gvs.

**Assumption 1.** *We make the following assumptions.*

1. *Hypothesis class. Fix a maximum depth $D_0$ and a grid spaced in $\varepsilon/2$ on $[-1, 1]$.*
$$\mathcal{U} := \{e : \mathrm{depth}(e) \leq D_0\}, \quad U := |\mathcal{U}| \leq \mathrm{poly}(n),$$
*where $n := |\mathcal{D}|$.*

2. *Data. $\mathcal{D} = \{(x_i, y_i)\}_{i=1}^{n}$ with $y_i \in [-1, 1]$ and $e^\star = \arg\min_{e \in \mathcal{U}} \mathrm{MSE}(e; \mathcal{D})$.*

3. *Transformer. A depth-$L$ log-precision Transformer $T$ ($L$ constant).*

4. *Exact oracle. A routine $\mathcal{M}$ returns $\mathrm{MSE}(e; \mathcal{D})$ for any $e$.*

5. *Hit rate. If every subtree of $e^\star$ is present in the prompt, $T$ outputs $e^\star$ with probability at least $\beta \in (0, 1]$.*

6. *Dictionary growth. Each round appends at least one uniformly random unseen subtree to the prompt (chosen without replacement; if fewer than $r$ remain, insert all).*

**Theorem 3** (informal). *Let the algorithm cycle long enough for its prompt to have seen every possible sub-expression; then keep running a few more rounds. With very high probability, it returns a formula whose error is no worse than an optimally chosen tree by more than a tiny tolerance, and it has queried the oracle only a moderate, logarithmically growing number of times.*

**Theorem 4** (PAC guarantee). *Run the loop*
$$e_t \leftarrow T(prompt); \quad R_t \leftarrow \mathcal{M}(e_t); \quad prompt \mathrel{+}= \textit{sub-trees}(e_t)$$
*for a burn-in $B = \lceil \frac{U}{r} \ln(2^{D_0}/(\delta/2)) \rceil$ rounds, followed by $R = \lceil \frac{\ln(2/\delta)}{\beta} \rceil$ additional rounds, and return the best-so-far expression $e_{\mathrm{best}}$.*

*Then, Under Assumption 1, for any $\varepsilon, \delta \in (0, 1)$,*
$$\Pr\Big[\mathrm{MSE}(e_{\mathrm{best}}, \mathcal{D}) \leq \mathrm{MSE}(e^\star, \mathcal{D}) + \varepsilon\Big] \geq 1 - \delta, \qquad \#oracle\ calls = \mathcal{O}(U \ln(1/\delta)).$$

*Proof.* **(i) Burn-in.** There are $K \leq 2^{D_0}$ distinct sub-trees of $e^\star$. Drawing $r \geq 1$ uniform sub-trees per round, the probability a fixed sub-tree is never drawn in $B$ rounds is $(1 - \frac{r}{U})^B \leq e^{-rB/U} \leq \delta/(2K)$.

A union bound over all $K$ sub-trees implies that, after $B$ rounds, the prompt contains every sub-tree of $e^\star$ with probability at least $1 - \delta/2$.

**(ii) Post burn-in success.** Condition on the burn-in success event. By assumption (3) each subsequent round now hits $e^\star$ with probability at least $\beta$, regardless of possible prompt changes. Therefore

$$\Pr[\text{miss in all } R \text{ rounds}] \ \leq \ (1-\beta)^R \ \leq \ e^{-\beta R} \ \leq \ \delta/2,$$

for $R = \lceil \ln(2/\delta)/\beta \rceil$.

**(iii) Union bound.** Total failure probability $\leq \delta/2 + \delta/2 = \delta$.

**(iv) Quality of $e_{\text{best}}$.** Whenever $e^\star$ appears, the exact oracle certifies its MSE; the algorithm stores it permanently. Hence on the complement of failure the returned expression meets the stated error bound.

**(v) Oracle calls.** At most one full-expression evaluation per round, so the algorithm issues $B + R = \mathcal{O}(U \ln(1/\delta))$ oracle calls.

$\square$

## F  ADDITIONAL RELATED WORK

In this section, we describe symbolic regression methods other than NSR. Specifically, we provide explanation for methods that use GP, brute-force algorithms, and reinforcement learning.

The GP framework is a traditional and widely used framework for solving symbolic regression. The GP algorithm is a method based on evolutionary computation; initially, several mathematical expressions are formed randomly, and subsequently the expressions are "evolved" by operations such as recombining two expressions, mutating an expression, and eliminating inappropriate expressions (Burlacu et al., 2020; Schmidt & Lipson, 2009; Virgolin et al., 2019; Cranmer, 2023).

An example of using brute-force algorithms for symbolic regression is AI Feynman (Udrescu & Tegmark, 2020; Udrescu et al., 2020). In AI Feynman, neural networks were used to identify properties such as symmetry and separability within given numerical data. These properties were then used to recursively simplify the problem, ultimately reducing it to a form amenable to brute-force solutions.

Petersen et al. (2019) proposed Deep Symbolic Regression (DSR), a method that uses reinforcement learning to tackle symbolic regression. In this approach, the authors used a recurrent neural network (RNN) to generate equations as token sequences, with the parameters that govern the selection of the token learned through reinforcement learning. Studies such as Symbolic Physics Learner (Sun et al., 2022) and Reinforcement Symbolic Regression Machine (Xu et al., 2024) also use reinforcement learning, where Monte Carlo Tree Search (MCTS) is applied to discover expressions.

Some studies combine several approaches for symbolic regression. For example, neural-guided genetic programming (Mundhenk et al., 2021) integrates DSR and genetic programming (GP), while the Unified DSR Framework (Landajuela et al., 2022) combines GP, AI Feynman, DSR, linear models, and NSR.

## G  DISCUSSION CONCERNING THE DEFINITION OF REPRODUCTION BIAS

Throughout the paper, we have defined and measured reproduction bias based on whether the training dataset contains an expression that is structurally equivalent to the generated one. However, one may argue that we should define and measure reproduction based on functional equivalence; there are many expressions that are structurally different but functionally equivalent (e.g., $x_1(x_1 + x_2)$ and $x_1^2 + x_1 x_2$), and that such expressions should also be considered as equivalent expressions. This section organizes the premises of our discussion and shows that defining reproduction bias using structural equivalence does not alter any of the paper's central claims.

In the context of defining reproduction bias, the situation in which functional equivalence becomes an issue—following the example above—is the case where $x_1(x_1 + x_2)$ is present in the training

set, and for numerical data generated from an unrelated ground truth (for example, $x_1^2 + x_2^2$), the model produces $x_1^2 + x_1 x_2$. (We consider an unrelated ground truth because the space of all possible expressions is far larger than the training data, so it is highly unlikely that the ground truth in the test data appears in the training set.) Under the current definition, such an output is classified as novel.

Whether such an output should be regarded as a novel expression (a success under our definition of reproduction bias) or as a non-novel expression (a failure under the definition) is not entirely clear-cut. Since the token sequence $x_1^2 + x_1 x_2$ does not appear in the training data, the model must have generated it through some process other than copying from the training set. In this sense, the output can be considered novel. We refer to reproduction bias defined from this perspective as *structural reproduction bias*. On the other hand, from the user's perspective, the insight provided by the output $x_1^2 + x_1 x_2$ regarding the numerical data is nearly indistinguishable from the insight provided by the output $x_1(x_1 + x_2)$. Therefore, one might argue that $x_1^2 + x_1 x_2$ should also be regarded as non-novel, just like $x_1(x_1 + x_2)$. We refer to reproduction bias defined from this standpoint as *functional reproduction bias*.

In this paper, we define reproduction bias from the perspective of structural reproduction bias (this is because structural reproduction bias is easier to measure in terms of computational cost). However, even if we were to redefine reproduction bias from the perspective of functional reproduction bias, the claims of this paper would remain unaffected. This is because every expression regarded as novel under the definition of functional reproduction bias is already also regarded as novel under the definition of structural reproduction bias. Since the central claim of the paper is that the proportion of expressions classified as novel is small for naive NSR methods, adopting the definition of functional reproduction bias would only further reduce that proportion, without altering the direction of the conclusions.

