# OpenReview forum: "Can Test-time Computation Mitigate Reproduction Bias in Neural Symbolic Regression?"
_ICLR.cc/2026/Conference — Submitted to ICLR 2026_

### Official Review · Reviewer_gm9E · 2025-10-28

**Soundness:** 3
**Presentation:** 2
**Contribution:** 2
**Rating:** 4
**Confidence:** 5

**Summary:**

This paper examines reproduction bias in Transformer-based symbolic regression, showing that models mainly memorize training expressions instead of composing new ones. It provides a theoretical justification and introduces NSR-GVS, a test-time algorithm that verifies and reuses accurate sub-expressions to reduce bias.

**Strengths:**

The main strength is that it explicitly demonstrates memorization behavior in neural symbolic regression.

**Weaknesses:**

The weakness is that these insights are not new in 2025, as several recent studies have reached similar conclusions.

**Questions:**

1. Since symbolic regression has been proven to be NP-hard [1], it is not surprising that neural symbolic regression struggles to find exact target equations. While the paper provides valuable empirical evidence, the results confirm known limitations rather than revealing new behavior.
2. The related work section is weak. Recent top-tier conferences (2024–2025) have largely moved away from end-to-end neural models like NeSymReS toward iterative or optimization-based symbolic regression methods, which align closely with the concept of “test-time computation” discussed in this paper. However, these recent works are barely acknowledged, suggesting the authors are not fully aware of current trends.
3. The experiments rely only on the Feynman benchmark. Additional tests on black-box datasets from SRBench would help validate the method’s effectiveness and generality.
4. The paper measures reproduction bias syntactically rather than semantically. This means equivalent equations in different algebraic forms may be incorrectly labeled as novel. A semantic-level comparison would provide a more accurate measure of reproduction bias.

[1] Virgolin, Marco, and Solon P. Pissis. "Symbolic Regression is NP-hard." Transactions on Machine Learning Research.

---

> ### Author Response · Authors · 2025-11-21
>
> Dear reviewer gm9E, thank you for taking the time to review our paper and sharing your insightful comments. We will address your questions and concerns below.
>
> ***
>
> >W1: The weakness is that these insights are not new in 2025, as several recent studies have reached similar conclusions.
>
> A1: The findings of this study, including the fact that NSR models cannot compositionally generate expressions while reasoning about accuracy and the identification of reproduction bias in neural symbolic regression (NSR), provide meaningful and sufficient novelty.
>
> For instance, with respect to reproduction bias, there has been little discussion on training-data memorization within NSR. Even in E2E [1] or transformer4sr [2], where memorization is mentioned to some extent, there is no observation pointing out that most expressions generated by the model are direct reproductions of those in the training set.
>
> With respect to the theoretical analysis, prior studies have not recognized that transformers not only fail to output the exact target equation, but also incapable of evaluating the accuracy of competing outputs. In this sense, our analysis can be viewed as an extension of prior theoretical work on symbolic regression [3] —such as studies identifying its NP-hardness—developed further within the context of auto-regressive expression generation.
>
> Please also refer to the response to Q1 since the points discussed there are also relevant.
>
> [1] Kamienny, Pierre-Alexandre, et al. "End-to-end symbolic regression with transformers." Advances in Neural Information Processing Systems 35 (2022): 10269-10281.
>
> [2] Lalande, Florian, et al. "A transformer model for symbolic regression towards scientific discovery." arXiv preprint arXiv:2312.04070 (2023).
>
> [3] Virgolin, Marco, and Solon P. Pissis. "Symbolic Regression is NP-hard." Transactions on Machine Learning Research.
>
> ***
>
> >Q1: Since symbolic regression has been proven to be NP-hard, it is not surprising that neural symbolic regression struggles to find exact target equations. While the paper provides valuable empirical evidence, the results confirm known limitations rather than revealing new behavior.
>
> A2: Both our theoretical and empirical analyses reveal previously unknown behaviors of Transformers in NSR. We demonstrate that Transformers suffer from a limitation that is more fundamental than simply failing to recover exact target equations; they **cannot distinguish between expressions based on numerical accuracy**, resulting in drawing expressions from the small space of expressions contained in the training data.
>
> First, as has been previously noted, the search space of full expressions is extremely large, making it difficult to find the exact target equation that perfectly fits the numerical data. However, even without identifying the exact target equation, there exist many “reasonably good” expressions that approximate the numerical data with high accuracy. Symbolic regressors capable of compositional generation of such expressions are therefore valuable.
>
> On the other hand, our theoretical analysis reveals that Transformers cannot compositionally generate expressions based on numerical accuracy; in other words, they are unable to distinguish between the target equation, “reasonably good” expressions, and entirely poor expressions.
>
> Moreover, the empirical analysis shows that a naive NSR model can essentially only generate expressions present in the training data. As a result, even “reasonably good” expressions cannot be produced unless they are included in the training set. These findings highlight limitations of NSR that have not been previously reported.
>
> In other words, there is a significant difference between merely “struggling to find exact target equations” and being unable to generate expressions compositionally while considering numerical accuracy, or being limited to producing only expressions present in the training data. Our contribution lies in pointing out that naive NSR methods have a more fundamental limitation beyond merely being unable to find exact target equations.

---

> > ### Author Response · Authors · 2025-11-21
> >
> > >Q2: The related work section is weak. Recent top-tier conferences (2024–2025) have largely moved away from end-to-end neural models like NeSymReS toward iterative or optimization-based symbolic regression methods, which align closely with the concept of “test-time computation” discussed in this paper. However, these recent works are barely acknowledged, suggesting the authors are not fully aware of current trends.
> >
> > A3: We have added the following to the Related Work section:
> >
> > *More recently, there has been growing research on methods that iteratively refine mathematical expressions, most of which rely on large language models (LLMs) rather than pre-trained Transformer models. These methods are similar to NSR-gvs, one of the test-time computation approaches considered in this paper, in that they iteratively improve their outputs by incorporating feedback from the generated expressions. In [1] and [2], the authors introduce approaches in which a base equation structure is generated using LLMs, and the equation is subsequently improved iteratively by receiving feedback from external numerical solvers. While [3] follows a similar methodology, it incorporates supplementary descriptions regarding the variables in the prompt, facilitating more effective use of the LLM’s scientific knowledge. [4] proposes a method that uses an LLM to identify “concepts” representing features of high-performing expressions and leverages them to further evolve a set of equations. [5] introduces an iterative algorithm that replaces features of suboptimal expressions at each step while incorporating relevant expressions as needed.*
> >
> > *Pre-training-free methods described above may have the potential to address some of the limitations of conventional NSR approaches. On the other hand, using a pre-trained Transformer, as opposed to an LLM, offers certain advantages similar to those of small language models (SLMs), such as keeping the model size manageable and enhancing domain specificity through careful design of the training data. For these reasons, we believe that conducting a deeper analysis of pre-trained Transformer-based NSR and exploring ways to improve it remains a valuable research direction.*
> >
> > [1] Merler, Matteo, et al. "In-context symbolic regression: Leveraging large language models for function discovery." arXiv preprint arXiv:2404.19094 (2024).
> >
> > [2] Sharlin, Samiha, and Tyler R. Josephson. "In context learning and reasoning for symbolic regression with large language models." arXiv preprint arXiv:2410.17448 (2024).
> >
> > [3] Shojaee, Parshin, et al. "Llm-sr: Scientific equation discovery via programming with large language models." arXiv preprint arXiv:2404.18400 (2024).
> >
> > [4] Grayeli, Arya, et al. "Symbolic regression with a learned concept library." Advances in Neural Information Processing Systems 37 (2024): 44678-44709.
> >
> > [5] Zhang, Hengzhe, et al. "RAG-SR: Retrieval-augmented generation for neural symbolic regression." The Thirteenth International Conference on Learning Representations. 2025.
> >
> > ***
> >
> > >Q3: The experiments rely only on the Feynman benchmark. Additional tests on black-box datasets from SRBench would help validate the method’s effectiveness and generality.
> >
> > A4: We plan to include the black-box datasets from SRBench in the experiments of Section D.2 and to present the results during the discussion period.
> >
> > ***
> >
> > >Q4: The paper measures reproduction bias syntactically rather than semantically. This means equivalent equations in different algebraic forms may be incorrectly labeled as novel. A semantic-level comparison would provide a more accurate measure of reproduction bias.
> >
> > A5: It is computationally impractical to verify semantic equivalence between a generated output and all expressions in the training data. Accordingly, we employ a syntactic-level comparison in this work.
> >
> > However, we would like to note that even if we were to measure reproduction bias by semantic-level comparison, the claims of this paper would remain unaffected. This is because every expression regarded as novel under semantic-level comparison is already also regarded as novel under syntactic-level comparison. Since the central claim of the paper is that the proportion of expressions classified as novel is small for naive NSR methods, adopting semantic-level comparison would only further reduce that proportion, without altering the direction of the conclusions.
> >
> > We have included the preceding discussion in Appendix G.

---

> > > ### Author Response · Authors · 2025-11-27
> > >
> > > Dear reviewer gm9E,
> > >
> > > We thank you for your invaluable contribution to improving our research.
> > > We have carefully considered each of your comments, and we believe that your concerns have been fully addressed (we plan to provide the additional experimental results on the black-box datasets by November 30). Should you have additional questions or comments, we would be happy to address them.

---

> ### Author Response · Authors · 2025-12-01
>
> Dear reviewer gm9E,
>
> We will provide the additional experimental results on the black-box datasets.
>
> Similar to the experiments in Section D.2, we compared the performance of NeSymReS and four different test-time compute methods using the black-box dataset from SRBench. We conducted our evaluation on the 35 tasks with fewer than five independent variables, as NeSymReS supports inputs with at most five independent variables. The results are shown in the table below (we do not report error bars for NSR-gvs + TPSR because experiments with multiple seeds have not yet been completed). We have also added corresponding plots to Section D.2 (please refer to Figure 7).
>
> Consistent with the results on the AI-Feynman dataset, the proposed method, NSR-gvs, improves performance, and combining it with TPSR leads to further gains. This demonstrates the effectiveness and generality of our method.
>
> |Method|NeSymReS (Beam Size = 5)|NeSymReS (Beam Size = 150)|NeSymReS + TPSR|NSR-gvs|NSR-gvs + TPSR|
> |:---:|:---:|:---:|:---:|:---:|:---:|
> |Expressions with $R^2$ > 0.5|**19.0** (±1.35)|35.2 (±1.35)|33.4 (±2.04)|**35.2** (±4.86)|**56.7**|
> |Expressions with $R^2$ > 0.75|**13.3** (±1.35)|20.0 (±0.0)|18.2 (±3.03)|**23.8** (±1.35)|**36.7**|
> |Expressions with $R^2$ > 0.9|**5.71** (±0.0)|5.71 (±0.0)|4.73 (±1.27)|**5.71** (±2.33)|**10.0**|
>
> Finally, thank you again for your thoughtful feedback, which has significantly helped us to improve our work.

---

### Official Review · Reviewer_B1hK · 2025-10-30

**Soundness:** 2
**Presentation:** 2
**Contribution:** 2
**Rating:** 2
**Confidence:** 4

**Summary:**

This paper presents a valuable empirical and theoretical investigation into a critical limitation of Neural Symbolic Regression (NSR) methods: ​reproduction bias, which is the tendency of models to generate expressions nearly identical to those seen during training. The authors compellingly demonstrate that under standard inference, models like NeSymReS struggle to produce expression structures not present in the training data, and this bias correlates with poor performance on such unseen expressions. A key contribution is the exploration of various test-time computation strategies (e.g., TPSR, NSR-gvs) to mitigate this bias. The paper also provides a theoretical argument, based on circuit complexity, suggesting Transformers may lack the inherent capacity for compositional generation guided by numerical data. The empirical findings are robust, showing that while these strategies can reduce reproduction bias, this reduction does not consistently lead to improved numerical accuracy, raising important questions for future research.

**Strengths:**

**Identification and Empirical Demonstration of a Critical Issue**:​​ The paper's greatest strength is its clear identification and systematic empirical demonstration of reproduction bias in NSR. The experimental design, using "not_included" and "baseline" datasets, effectively isolates the problem and shows its severe impact on generalization.

**Rigorous Exploration of Test-Time Strategies**:​​ The paper goes beyond merely identifying the problem by rigorously evaluating multiple test-time computation strategies (TPSR, NSR-gvs, and their combination). The analysis in Figures 4 and 7, which examines the trade-off between reproduction bias, numerical accuracy, and computational cost, is thorough and provides valuable insights for the community. The finding that bias reduction and accuracy improvement are not directly correlated is a nuanced and important result.

**Theoretical Ambition**:​​ Attempting to ground the limitations of NSR in computational complexity theory is a commendable and ambitious goal. The argument that the "last-token prediction" problem may be too hard for bounded-precision Transformers adds a theoretical layer to the empirical observations.

**Weaknesses:**

​**Fundamental Flaw in the Definition and Evidence for Reproduction Bias**:​​ The central concept of "reproduction bias" is potentially flawed. The criterion for bias—whether a generated expression tree structure is present in the training data—does not account for ​symbolic equivalence. Many expressions are functionally identical (e.g., x1*(x1+x2)and x1**2 + x1*x2). Therefore, a model generating an equivalent but syntactically different expression should be considered a success, not evidence of bias. The provided evidence may actually indicate that larger models (e.g., transformer4sr trained on 1.5M samples) are betterat finding valid, equivalent expressions, explaining their higher rate of "novel" (i.e., non-identical) generations (6%) compared to a smaller model (NeSymReS on 100K samples, <1%). The paper would be significantly strengthened by evaluating whether generated expressions are functionally equivalent to the ground truth, not just structurally identical.

**Limited Scope and Questionable Generalizability of Findings**:​​ The empirical analysis relies heavily on models trained with relatively small datasets (100K expressions for the main analysis). The performance of state-of-the-art NSR models, which are pre-trained on orders of magnitude more data (e.g., 100M expressions for the official NeSymReS model), is not investigated. It remains an open question whether the severe reproduction bias demonstrated here is merely a symptom of under-training or a fundamental limitation that persists even at scale. Testing on the official pre-trained models is crucial to determine the broader relevance of the findings.

**Theoretical Argument is Misaligned with the Practical Goal of SR**:​​ The theoretical analysis in Section 4, while interesting, may be misleading. The "last-token prediction" task is an artificial construct. The ultimate goal of symbolic regression is to find anyexpression that fits the data well, regardless of its specific token sequence. A model might predict the "wrong" final token according to a specific canonical form but still produce a functionally equivalent and highly accurate expression. Therefore, the inability to solve this specific task does not necessarily prove that Transformers cannot be effective for the broader, more flexible goal of symbolic regression. The argument would be more compelling if it addressed the problem of discovering equivalent expressions.

**Questions:**

None.

---

> ### Author Response · Authors · 2025-11-21
>
> Dear reviewer B1hk, thank you for taking the time to review our paper and sharing your insightful comments. We will address your concerns below.
>
> ***
>
> >W1: **Fundamental Flaw in the Definition and Evidence for Reproduction Bias**:​​ The central concept of "reproduction bias" is potentially flawed. The criterion for bias—whether a generated expression tree structure is present in the training data—does not account for ​symbolic equivalence. Many expressions are functionally identical (e.g., x1*(x1+x2) and x1*2 + x1x2). Therefore, a model generating an equivalent but syntactically different expression should be considered a success, not evidence of bias. The provided evidence may actually indicate that larger models (e.g., transformer4sr trained on 1.5M samples) are better at finding valid, equivalent expressions, explaining their higher rate of "novel" (i.e., non-identical) generations (6%) compared to a smaller model (NeSymReS on 100K samples, <1%). The paper would be significantly strengthened by evaluating whether generated expressions are functionally equivalent to the ground truth, not just structurally identical.
>
> A1: While we see no fundamental flaw in our definition of reproduction bias, we would like to provide additional clarification regarding functional equivalence. Our key point is that whether one adopts functional equivalence or structural equivalence, the claims of the paper remain unchanged.
>
> We first clarify that we are not examining whether the generated expressions are equivalent to the **ground-truth equation**. In our current evaluation of reproduction bias, we are examining whether the generated expressions are **equivalent to any expression in the training set**. If no such structurally identical expression is found, we regard the output as a novel expression.
>
> Therefore, in the context of defining reproduction bias, the situation in which functional equivalence becomes an issue—following the example above—is the case where x1*(x1+x2) is present in the training set, and for numerical data generated from an unrelated ground truth (for example, x1^2 + x2^2), the model produces x1^2 + x1x2​. (We consider an unrelated ground truth because the space of all possible expressions is far larger than the training data, so it is highly unlikely that the ground truth in the test data appears in the training set.) Under the current definition, such an output is classified as novel.
>
> Whether such an output should be regarded as a novel expression (a success under our definition of reproduction bias) or as a non-novel expression (a failure under the definition) is not entirely clear-cut. Since the token sequence x1^2 + x1x2​ does not appear in the training data, the model must have generated it through some process other than copying from the training set. In this sense, the output can be considered novel. We refer to reproduction bias defined from this perspective as *structural reproduction bias*.
>
> On the other hand, from the user’s perspective, the insight provided by the output x1^2 + x1x2 regarding the numerical data is nearly indistinguishable from the insight provided by the output x1*(x1+x2)​. Therefore, one might argue that x1^2 + x1x2 should also be regarded as non-novel, just like x1*(x1+x2). We refer to reproduction bias defined from this standpoint as *functional reproduction bias*.
>
> In this paper, we define reproduction bias from the perspective of structural reproduction bias (this is because structural reproduction bias is easier to measure in terms of computational cost). However, even if we were to redefine reproduction bias from the perspective of functional reproduction bias, the claims of this paper would remain unaffected. This is because every expression regarded as novel under the definition of functional reproduction bias is already also regarded as novel under the definition of structural reproduction bias. Since the central claim of the paper is that the proportion of expressions classified as novel is small for naive NSR methods, adopting the definition of functional reproduction bias would only further reduce that proportion, without altering the direction of the conclusions.
>
> From the discussion above, we conclude that the definition of reproduction bias does not suffer from any fundamental problems.
> We have included the preceding discussion in Appendix G.

---

> > ### Author Response · Authors · 2025-11-21
> >
> > >W2:  **Limited Scope and Questionable Generalizability of Findings**:​​ The empirical analysis relies heavily on models trained with relatively small datasets (100K expressions for the main analysis). The performance of state-of-the-art NSR models, which are pre-trained on orders of magnitude more data (e.g., 100M expressions for the official NeSymReS model), is not investigated. It remains an open question whether the severe reproduction bias demonstrated here is merely a symptom of under-training or a fundamental limitation that persists even at scale. Testing on the official pre-trained models is crucial to determine the broader relevance of the findings.
> >
> > A2: The upper limit of 1.5M equations for the training dataset size was chosen in accordance with the practical setting used in transformer4sr [1]. For the analysis on the NeSymReS model, the upper limit of 100K expressions was chosen due to the complicated nature of training data generation in the NeSymReS method; since NeSymReS generates training data dynamically during training time, it is computationally expensive to check whether a generated expression was fed to the model afterwards.
> > Due to the high computational cost, it is not feasible to conduct experiments on a training dataset of 100M samples.
> >
> > Additionally, the following two pieces of evidence can be presented to suggest that reproduction bias may also exist in larger training datasets:
> > - As shown in Figure 5 of Appendix D, reproduction bias does not improve even when the size of the training dataset is varied.
> > - in terms of the ratio between the training data size and the search space, the setting in Section 5.1 employs a larger effective dataset than the prior study that trained on 100M expressions. This is because we also restricted the number of operators in Section 5.1. This makes it unlikely that the observed reproduction bias is attributable to under-training.
> >
> > [1] Lalande, Florian, et al. "A transformer model for symbolic regression towards scientific discovery." arXiv preprint arXiv:2312.04070 (2023).
> >
> > ***
> >
> > >W3: **Theoretical Argument is Misaligned with the Practical Goal of SR**:​​ The theoretical analysis in Section 4, while interesting, may be misleading. The "last-token prediction" task is an artificial construct. The ultimate goal of symbolic regression is to find any expression that fits the data well, regardless of its specific token sequence. A model might predict the "wrong" final token according to a specific canonical form but still produce a functionally equivalent and highly accurate expression. Therefore, the inability to solve this specific task does not necessarily prove that Transformers cannot be effective for the broader, more flexible goal of symbolic regression. The argument would be more compelling if it addressed the problem of discovering equivalent expressions.
> >
> > A3: The “last-token prediction” task is not an entirely artificial construct; it represents a real challenge that a transformer actually faces when generating the final token during the auto-regressive process of equation generation. In this task, **the issue of equivalent expressions is also taken into account**.
> >
> > In the “last-token prediction” task, the model is required to complete an equation that is complete except for one remaining token, by selecting the most suitable leaf token in terms of numerical accuracy. Therefore, no “canonical form” exists from the beginning, and if the completion of two different leaf tokens both result in highly accurate expressions, either token can be considered correct.

---

> > > ### Author Response · Authors · 2025-11-27
> > >
> > > Dear reviewer B1hk,
> > >
> > > We appreciate your extensive review and constructive feedback.
> > > We believe that our response above has fully addressed your concerns, particularly those related to the functional equivalence of expressions.
> > > Should you have additional questions or comments, we would be happy to address them.

---

### Official Review · Reviewer_ELR6 · 2025-10-31

**Soundness:** 3
**Presentation:** 3
**Contribution:** 2
**Rating:** 6
**Confidence:** 5

**Summary:**

The paper studies why neural symbolic regression (NSR) models often fail to produce novel equations and instead copy training expressions (a "reproduction bias"). Theoretically, under the circuit-complexity assumption TC0 ̸= NC1, and log-precision constraints, the authors prove that for sufficiently large problem sizes no bounded-depth, polynomial-width Transformer can even solve a last-token prediction task (i.e. predict the last leaf node of otherwise complete expression that minimizes  the numeric data), implying difficulty with compositional generation  while accounting for numerical loss.

Empirically, they show that in a 1.5M training dataset, most decoded expressions are copies of the training set and accuracies decreases significantly when the target tree is unseen.

Finally they then test three test-time strategies: larger beam sizes, TPSR (MCTS), and their proposed NSR-gvs (verified subtrees). Methods that inject extra information increase novelty but don’t reliably improve accuracy, while simply enlarging the beam often boosts accuracy without reducing reproduction bias,revealing a tension between escaping memorization and maintaining fit.

**Strengths:**

- The identified reproduction bias is significant and analyzed soundly from both theoretical and empirical perspective

**Weaknesses:**

- While the paper explores different strategies and approaches, no proposed solution, as acknowledged by the author in the conclusions, seems to both decrease the reproduction bias and increase accuracy.
- Experiments are carried out with a maximum dataset size of 1.5M equations, whereas established work in the field trains these models with up to 100M. Hence, the conclusions might be biased by this difference in scale.
- The connection between the theoretical and empirical evaluation settings is poorly described.

**Questions:**

Feel free to address the weakness I have mentioned

---

> ### Author Response · Authors · 2025-11-21
>
> Dear reviewer ELR6, thank you for taking the time to review our paper and sharing your insightful comments. We will address your questions and concerns below.
>
> ***
>
> >W1: While the paper explores different strategies and approaches, no proposed solution, as acknowledged by the author in the conclusions, seems to both decrease the reproduction bias and increase accuracy.
>
> A1: Our point was specifically tied to the phrase “to a significant extent.”
>
> *“The main limitation of this work is the absence of a method that simultaneously mitigates reproduction bias and improves numerical accuracy **to a significant extent**.”*
>
> The intention behind our original statement was not to imply that the paper proposes no method capable of reducing reproduction bias or improving accuracy at all, but rather that—based on the results presented—none of the explored approaches appears to achieve both goals to a significant extent simultaneously.
>
> We hope this resolves the misunderstanding, and we appreciate the opportunity to clarify.
>
> ***
>
> >W2: Experiments are carried out with a maximum dataset size of 1.5M equations, whereas established work in the field trains these models with up to 100M. Hence, the conclusions might be biased by this difference in scale.
>
> A2: The upper limit of 1.5M equations for the training dataset size was chosen in accordance with the practical setting used in transformer4sr [1].
> Due to the high computational cost, it is not feasible to conduct experiments on a training dataset of 100M samples.
>
> Additionally, the following two pieces of evidence can be presented to suggest that reproduction bias may also exist in larger training datasets:
> - As shown in Figure 5 of Appendix D, reproduction bias does not improve even when the size of the training dataset is varied.
> - in terms of the ratio between the training data size and the search space, the setting in Section 5.1 employs a larger effective dataset than the prior study that trained on 100M expressions. This is because we also restricted the number of operators in Section 5.1. This makes it unlikely that the observed reproduction bias is attributable to under-training.
>
> [1] Lalande, Florian, et al. "A transformer model for symbolic regression towards scientific discovery." arXiv preprint arXiv:2312.04070 (2023).
>
> ***
>
> > W3: The connection between the theoretical and empirical evaluation settings is poorly described.
>
> A3: To clarify the relationship between the theoretical and empirical evaluations, we made the following revisions to lines 249~252 in Section 5.
>
>  - **Before**: *In Sec. 4, our theoretical analysis showed that Transformers lack the ability to generate expressions in a compositional manner while accounting for numerical data. Given the limitations of Transformers described above, this section empirically analyzes how expressions are actually generated by NSR models.*
>
> - **After**: *When generating expressions in an auto-regressive manner, a seemingly appropriate strategy would be to compositionally produce the next token that maximizes accuracy, conditioned on both the previously generated partial expression and the numerical data. However, our theoretical analysis from the last section showed that Transformers lack the ability to do so, bringing us to the following question:* How, in practice, does a transformer generate expressions during inference? *In this section, we empirically analyze how expressions are actually generated by NSR models.*
>
>
> To clarify further, the connection between the theoretical and empirical evaluations is as follows.
>
> First, when generating expressions in an auto-regressive manner, a seemingly appropriate strategy would be to compositionally produce the next token that maximizes accuracy, conditioned on both the previously generated partial expression and the numerical data.
>
> However, our theoretical evaluation shows that transformers do not have such an ability to compositionally generate expressions while taking numerical accuracy into account. This gives rise to the following question: *How, in practice, does a transformer generate expressions during inference?*
>
> We therefore formulated the hypothesis that, in practice, the transformer simply reproduces expressions present in the training data, and we evaluated this hypothesis under empirical conditions.

---

> > ### Author Response · Authors · 2025-11-27
> >
> > Dear reviewer ELR6,
> >
> > Thank you for your great efforts to review our paper.
> > We believe that our response above has fully addressed your concerns, and we hope that it further strengthens your confidence in our work.
> > If you have any further questions or feedback, please feel free to let us know.

---

### Author Response · Authors · 2025-12-03

Dear Area Chair and Reviewers,

We would first like to thank all reviewers for the significant time and effort they devoted to evaluating our paper and for offering insightful comments that strengthened the work. We have uploaded a revised version of the paper, with the newly added sections highlighted in blue.

We also thank the Area Chair for taking the time to carefully assess both our submission and our rebuttal despite the unexpected situation.

The following provides a concise summary of the reviewers’ key questions and concerns, along with the corresponding points in our rebuttal, to assist the Area Chair. Unfortunately, the discussion phase ended unexpectedly, potentially preventing reviewers from responding to our rebuttal. Nevertheless, we are confident that we addressed every question and fully resolved all concerns identified as weaknesses. Therefore, we believe that, had the discussion proceeded as intended, the initial scores would have improved.

***

>1. Reviewers B1hk and gm9E raised concerns regarding our use of structural rather than semantic equivalence between expressions when defining and measuring reproduction bias.

We clarified that even if reproduction bias were defined using semantic equivalence, **our claims would remain unaffected**, because the structural-equivalence–based measurement we adopt constitutes a stricter criterion.

***

>2. Reviewers B1hk and ELR6 raised concerns that the training dataset employed in our measurement of reproduction bias is relatively small (up to 1.5M expressions), whereas some existing models are trained on more than 100M expressions. Consequently, they questioned whether reproduction bias remains present when models are trained on much larger datasets.

We first clarified that the upper limit of 1.5M expressions in our training dataset **follows the practical experimental setup** used in prior work [1], thereby justifying our choice, and that it is computationally infeasible for us to conduct the same experiment on a dataset with 100M expressions.
We further argued that reproduction bias is very likely to remain even with substantially larger datasets, because:
 - Figure 5 in Appendix D shows that **the benefit of enlarging the training set quickly plateaus**.
 - By also constraining the search space in Section 5.1, we ensure that the relative effective dataset size exceeds that of the 100M-expression method, indicating that **our findings are not a consequence of under-training**.

[1] Lalande, Florian, et al. "A transformer model for symbolic regression towards scientific discovery." arXiv preprint arXiv:2312.04070 (2023).

***

>3. Reviewer gm9E raised doubts about the novelty of our theoretical analysis, referring to [2], which shows that finding exact target equations is NP-hard in symbolic regression.

We responded by emphasizing that **our analysis goes substantially beyond this known difficulty**: NSR models fail to assess the relative numerical accuracy of expressions altogether, a limitation that is strictly stronger than merely being unable to locate the exact target equation. This provides clear novelty beyond prior work.

[2] Virgolin, Marco, and Solon P. Pissis. "Symbolic Regression is NP-hard." Transactions on Machine Learning Research.

***

>4. Reviewer B1hk expressed concern that our theoretical analysis does not account for semantic equivalence when evaluating expressions generated by the Transformer.

We clarified that this concern stems from a misunderstanding of the theorem: the theorem evaluates expressions based on their numerical accuracy, which inherently incorporates semantic equivalence between expressions. Thus, semantic equivalence is indeed taken into account.

***

>5. Reviewer ELR6 identified as a weakness that the paper does not propose any method that simultaneously mitigates reproduction bias and improves numerical accuracy.

We clarified that this was due to an oversight: as stated in the main paper, our proposed method **does** both alleviate reproduction bias and enhance numerical accuracy.

***

In addition, we followed the reviewers’ suggestions by conducting further experiments and refining the manuscript accordingly.

Thank you again for your time and consideration.

Best regards,

The Authors

---

### Meta-Review · Area_Chair_7pq6 · 2026-01-07

**Summary:**

Reviewers broadly agree the paper convincingly surfaces an important failure mode (reproduction bias) and provides thoughtful analysis, but raise major concerns about (i) evaluation validity (syntactic vs semantic bias, missing equivalence checks), (ii) insufficient comparisons and limited benchmark breadth, (iii) unclear generalization at realistic pretraining scales, (iv) weak/dated related work positioning, and (v) a theory component that may not tightly support the practical SR claims. Collectively, these issues prevent a strong acceptance case in the current form.

**Reviewer Concerns:**

Two major concerns remain insufficiently addressed:

1. The paper focuses on structural (syntactic) reproduction bias, while two reviewers (ELR6, gm9E) emphasize the need to consider functional (semantic) reproduction bias. Although the authors argue that semantic bias is a subset of structural bias and thus does not affect their claims, I agree with the reviewers. In particular, the paper claims that reducing reproduction bias does not necessarily correlate with improved accuracy—but this conclusion may change if the analysis targets semantic reproduction bias. In other words, what if reducing semantic bias does correlate with accuracy?

2. Reviewer gm9E also points out that the paper misses or under-emphasizes 2024–2025 trends in symbolic regression toward iterative/optimization-based methods and broader notions of test-time compute. I agree this is a significant weakness in positioning and context.

**Reviewer Scores:**

I think none of the reviewers would change their ratings.

---

### Decision · Program_Chairs · 2026-01-26

Reject